# High-affinity peptides developed against calprotectin and their application as synthetic ligands in diagnostic assays

Cristina Díaz-Perlas [1], Benjamin Ricken [2], Lluc Farrera-Soler [1], Dmitrii Guschin[2], Florence Pojer [3], Kelvin Lau [3], Christian-Benedikt Gerhold [2] ✉ & Christian Heinis [1] ✉

Common inflammatory disorders such as ulcerative colitis and Crohn's disease are non-invasively diagnosed or monitored by the biomarker calprotectin. However, current quantitative tests for calprotectin are antibody-based and vary depending on the type of antibody and assay used. Additionally, the binding epitopes of applied antibodies are not characterized by structures and for most antibodies it is unclear if they detect calprotectin dimer, tetramer, or both. Herein, we develop calprotectin ligands based on peptides, that offer advantages such as homogenous chemical composition, heat-stability, site-directed immobilization, and chemical synthesis at high purity and at low cost. By screening a 100-billion peptide phage display library against calprotectin, we identified a high-affinity peptide ($K_d = 26 \pm 3$ nM) that binds to a large surface region (951 Å²) as shown by X-ray structure analysis. The peptide uniquely binds the calprotectin tetramer, which enabled robust and sensitive quantification of a defined species of calprotectin by ELISA and lateral flow assays in patient samples, and thus offers an ideal affinity reagent for next-generation inflammatory disease diagnostic assays.

Inflammatory bowel diseases (IBD), including Crohn's disease and ulcerative colitis, are characterized by chronic remittent episodes of inflammation in and beyond the GI tract that afflict individuals with substantial morbidity[1,2]. Since the identification and characterization of calprotectin in the 1980s, fecal calprotectin emerged as a non-invasive biomarker that discriminates between inflammatory and non-inflammatory diseases of the gut and portraits the disease course of human IBD. Calprotectin is expressed as a heterodimeric protein (Mrp-8/Mrp-14 or S100A8/S100A9) in neutrophils and can reach up to 50% of the soluble protein content in the cytoplasm. During intestinal inflammation, neutrophils migrate to the intestinal mucosa and release calprotectin into the lumen of the intestine, where high calcium ion concentrations are likely to lead to the formation of a tetrameric structure[3]. The quantity of the marker calprotectin detected in feces shows a good correlation with endoscopic and histological data and indicates the severity of IBD[1]. In healthy adults, median levels of fecal calprotectin are 10–34 µg/g; values between 80–160 µg/g can represent a mild organic disease or inflammation, and calprotectin above 160 µg/g is highly indicative of active inflammation in the gut, wherein the values for the categorizations vary slightly in literature[4–6]. As such, the marker is broadly used in the diagnosis of IBD and to monitor disease activity, response to therapy, and disease recurrence. Calprotectin can also be elevated in blood and may be used for monitoring other inflammatory diseases such as rheumatoid arthritis, cystic fibrosis, acute coronary syndrome, or sepsis[7]. Quantification of specific oligomeric states or metabolic products of calprotectin, as observed in blood, might offer additional value for the diagnosis of diseases[8–10].

[1]Institute of Chemical Sciences and Engineering, School of Basic Sciences, École Polytechnique Fédérale de Lausanne (EPFL), CH-1015 Lausanne, Switzerland. [2]BÜHLMANN Laboratories AG, Baselstrasse 55, CH-4124 Schönenbuch, Switzerland. [3]Protein Production and Structure Core Facility, School of Life Sciences, Ecole Polytechnique Fédérale de Lausanne (EPFL), Lausanne, Switzerland. ✉e-mail: cbg@buhlmannlabs.ch; christian.heinis@epfl.ch

Calprotectin is detected and quantified by assays that depend on antibodies as affinity reagents, such as ELISA, lateral flow assays (LFA), chemiluminescence immunoassays (CLIA), and turbidimetric assays. While the mono- or polyclonal antibodies used in such assays allow for sensitive and quantitative detection of calprotectin, side-by-side comparisons of immunoassays differ in reported calprotectin concentrations up to fivefold[11,12]. The observed variations may stem from heterogeneous standardization across assays from different providers but are likely also caused by general limitations of antibodies, such as protein heterogeneity, inter-batch variations, temperature-induced denaturation, and challenges in site-selective antibody immobilization[13–15]. Furthermore, calprotectin can occur as dimer or tetramer, depending on the sample type and treatment, and some antibodies recognize the different species to different extents. Binding studies with the various forms of calprotectin or epitope mapping by phage display have shed light on the binding sites of some calprotectin antibodies[16], but none have characterized the structure of the antibody-antigen complex. Thus, many of the commercially applied assays cannot specify whether calprotectin is detected as a dimer, tetramer, or both, which may be one of the main sources of variability between assay manufacturers.

In this study, we develop a calprotectin LFA based on peptide affinity reagents. We reasoned that peptides should offer advantages such as a homogenous product due to chemical synthesis, high stability for a prolonged shelf-life, site-directed immobilization, and low batch-to-batch variation. While antibodies are usually chosen as affinity reagents in diagnostic assays due to their low nanomolar or picomolar binding affinities, we speculated that such high affinities would not be needed due to the relatively high concentrations of calprotectin in patient samples. Peptides binding to targets of interest can be efficiently developed with in vitro evolution methods such as phage display and mRNA display[17] and typically display nanomolar affinities, which would be sufficiently strong for quantifying the typically high nanomolar calprotectin concentrations. Herein, we have screened the largest ever reported peptide phage display library, comprising over 100 billion different peptides[18], and isolated nanomolar binders to calprotectin. We characterized the binding epitope of the peptides by solving an X-ray structure of the calprotectin:peptide, and found that the peptides bind exclusively to tetrameric calprotectin and, thus, a defined species of the biomarker. Overall, our work here shows that peptides are suitable for the development of sensitive LFAs amenable to the quantification of calprotectin in complex biological samples.

## Results

### Phage-selected peptides have similar sequences

To develop the calprotectin-binding peptides, we performed phage display selections using three libraries comprising cysteine-rich peptides that are cyclized post-display into mono- or bicyclic peptides by chemically bridging pairs of cysteines using bis-electrophilic reagents (Fig. 1a)[19]. Libraries 1 and 2 were developed previously[18,20], with Library 2 being the largest peptide phage display reported so far (over 100 billion different monocyclic and bicyclic peptide sequences). Library 3 was newly cloned (Supplementary Fig. 1). To serve as the target, we biotinylated human calprotectin at random lysines (biotin-NHS) and immobilized it on magnetic streptavidin or neutravidin beads, alternating the two bead types to prevent enrichment of binders to one of the two proteins. We screened against recombinantly expressed calprotectin B-RCAL (BÜHLMANN recombinant calprotectin; $His_6$-linker-S100A9-linker-S100A8) that fused the S100A9 and S100A8 proteins via a cleavable linker (LEVLFQ/GP; all eight amino acids of the linker belong to the HRV 3C protease recognition site; cleavage site indicated) and that carried an N-terminal His-tag appended via a second cleavable linker (MM[H]$_6$LEVLFQ/GP)[21]. Because B-RCAL has the same structure and essentially the same immunological properties as native calprotectin purified from granulocytes[21], we used the recombinant

protein without removing the His-tag or cleaving the 8-amino acid linker between the two proteins for all phage display selections and the initial peptide characterization experiments. To screen for the tetramer, anticipated to be the form found in blood and serum patient samples[22,23], we performed all selection steps in the presence of calcium chloride so that two B-RCAL heterodimeric fusion proteins formed a tetrameric structure.

To isolate target-specific peptide sequences, we initially performed three phage display selection rounds. After the second or third round, the number of captured phage strongly increased for all libraries, indicating enrichment of target-specific binders (Supplementary Fig. 2). DNA sequencing revealed convergence to sequences sharing the common peptide motif PL$^F/_Y$ (Fig. 1b for Library 2 and Supplementary Fig. 3 for Library 1 and 3), which pointed further to the enrichment of target-specific binders. The selection results were best for the largest library (Library 2), where we found many different peptides, all carrying the consensus motif. Overall, the peptides could be grouped into families with similar but slightly different consensus sequences extended from the PL$^F/_Y$ motif that appeared to be influenced to some extent by the type of linker used for peptide library cyclization (Fig. 1b). For example, peptides isolated with linker **3** shared the extended consensus sequence FP$^L/_M$$^F/_Y$.

To identify the peptides with the highest affinities, we performed two additional selection rounds using less target protein and thus increasing the selection pressure (Round 5; Fig. 1b). We selected several of the peptides isolated from these later rounds, with a focus on those isolated from Library 2. We synthesized the peptides with an N-terminal fluorescein label, cyclized random pairs of reduced cysteines by adding an excess of cyclization reagents **1** to **6**, and chromatographically separated the three isomers formed by bridging different cysteine pairs (Supplementary Fig. 4). We next measured their binding to calprotectin by fluorescence polarization (FP). For each peptide sequence, typically, one of the three isomers produced bound much better than the others, with dissociation constants in the nanomolar range (Fig. 1b and Supplementary Figs. 3, 5). We expected to see the preferential binding of one isomer, as the differential bridging of cysteine pairs results in completely different peptide topologies[19]. Regardless, the nanomolar binding affinity was confirmed by surface plasmon resonance (SPR) as an orthogonal binding assay, wherein the affinity measured by SPR was substantially weaker for some of the peptides (Supplementary Fig. 6a). In all assays, binding was measured in the presence of calcium chloride (2 mM) and thus to the tetrameric form of calprotectin. In the absence of calcium ions, weak or no binding was observed, suggesting that the peptides bind to a region spanning the two heterodimers. Analysis by dynamic light scattering (DLS) and mass photometry (MP), that measure the hydrodynamic radius and mass of proteins, respectively, assured that the calprotectin used indeed occurred as tetrameric structure in the presence of calcium ions, and as a dimer in the absence of calcium ions (Supplementary Fig. 7).

### Peptides bind in "dumbbell" and linear form

Of all peptides binding with nanomolar affinity, peptide 3 could be most easily measured due to a particularly strong FP increase upon binding to calprotectin (around 50%; Supplementary Fig. 5), and we thus focused on this peptide for further characterizations. We synthesized the three regioisomers of peptide 3 following a previously established strategy based on orthogonal cysteine side chain protecting groups[19]. FP measurement showed that the "dumbbell" shaped isomer 1 was the main calprotectin binder ($K_d = 24 \pm 10$ nM; Fig. 2a), which bridged the cysteine pairs Cys1-Cys2 and Cys3-Cys4 (chemical linker **3**). The other two isomers showed only a small FP increase (10–15%) that indicated no or partial binding (Fig. 2a, isomer 2 and 3). Due to the presence of two identical calprotectin proteins forming a 2-fold symmetry, each calprotectin tetramer can bind two peptides (or each heterodimer one peptide). Thus, we calculated the dissociation

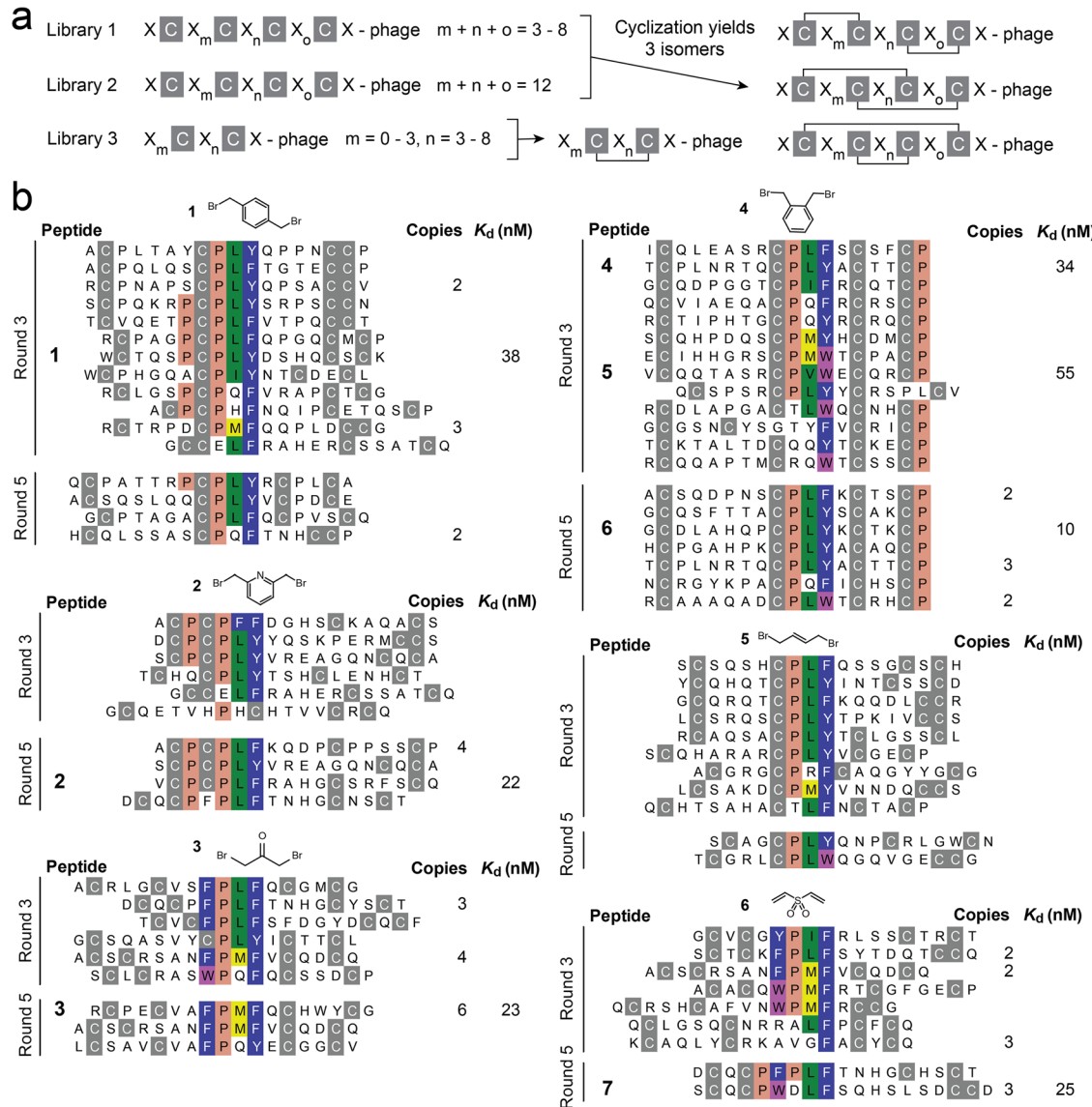

**Fig. 1 | Phage display selection of (bi)cyclic peptides against human calprotectin tetramer. a** Formats of the phage display peptide libraries. C cysteine, X any of the 20 canonical amino acids. Prior to selection, pairs of cysteines were bridged using the chemical reagents shown above the peptide sequences in (**b**). **b** Sequences isolated after three and five rounds of phage display selection from Library 2. Results from Library 1 and 3 are shown in Supplementary Fig. 3. Sequence similarities are colored, and the abundance of each peptide is indicated. Binding affinities are shown for peptides that were synthesized, and that had a signal measurable by FP. The indicated $K_d$ values represent binding to calprotectin in the presence of calcium ions, and thus the tetrameric form, using the concentration of heterodimeric B-RCAL protein in the assay rather than the tetramer concentration, as each calprotectin tetramer binds two peptides. The indicated $K_d$ values represent the mean of three independent measurements.

constants using a 1:1 heterodimer:peptide ratio to better reflect the affinity, which gives larger numbers and apparently weaker binding constants. For this reason, all FP binding curves display the concentration of the heterodimer fusion protein B-RCAL and not the concentration of tetramer. SPR analysis of the three isomers confirmed that peptide 3 binds in the dumbbell form and with nanomolar affinity, though with a twofold weaker dissociation constant than found in the FP assay ($K_d = 51$ nM by SPR versus 24 nM by FP; Supplementary Fig. 6b). We did not find out why the affinity measurements by SPR gave weaker affinities, but could imagine that it is due the binding measurement in solution (FP) versus on a surface (SPR).

We next assessed whether only one chemical bridge in peptide 3 would be sufficient for binding, which would simplify the synthesis compared to double-bridged peptides that require orthogonal thiol-protecting groups and sequential deprotection and cyclization. Peptide 3 with Cys1-Cys2 bridged (isomer 1a) and peptide 3 with Cys3-Cys4

bridged (isomer 1b) both bound as well as the double-bridged peptide 3 (Fig. 2b and Supplementary Fig. 8). We subsequently tested the linear peptide 3 (all four cysteines replaced with serines) and found that it also binds with comparable affinity ($K_d = 26 \pm 3$ nM; Fig. 2b). This result was unexpected, as we have performed selections with bicyclic peptides against more than twenty targets over the last ten years, and the (bi)cyclic peptides have always shown better binding affinities than their linear analogs[24]. While cyclic peptides are preferred in therapeutic applications due to their higher resistance towards plasma proteases, finding a linear peptide ligand against calprotectin as a detection agent was a positive outcome due to the ease of synthesis and immobilization.

## Peptide binds in elongated conformation to tetramer

To understand which residues of linear peptide 3 are most important for binding, we performed an alanine scan. Mutation to alanine led

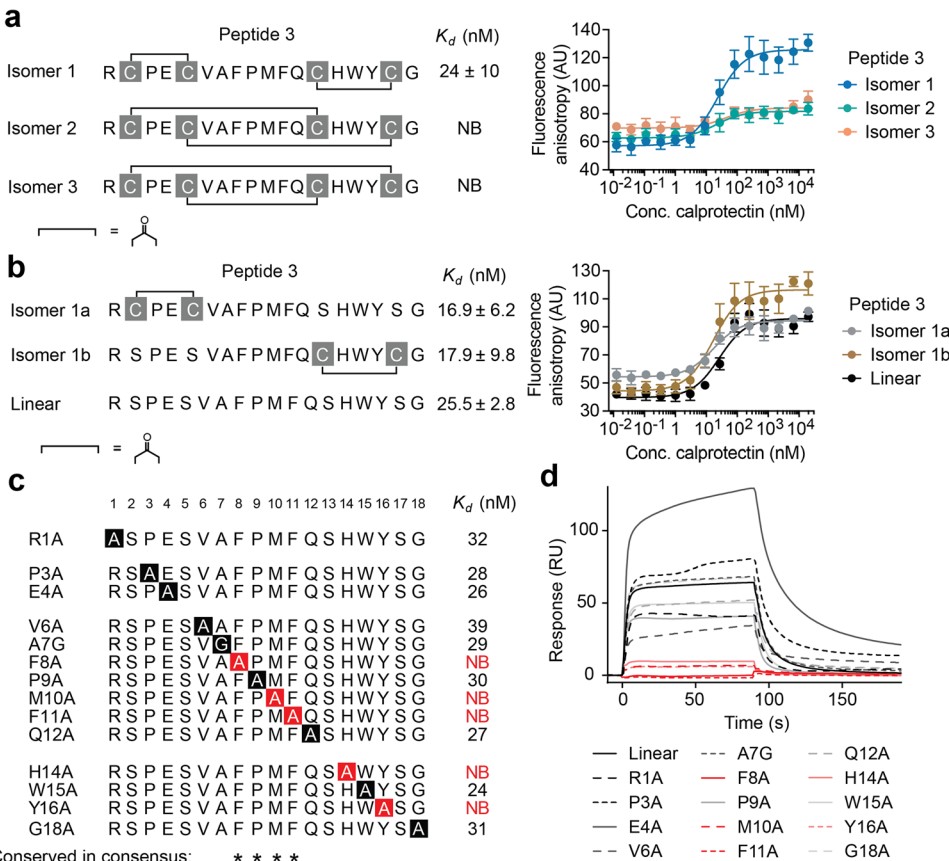

**Fig. 2 | Characterization of peptide 3. a** FP measurements of binding of peptide 3 isomers to calprotectin tetramer (in the presence of calcium ions), indicated as the concentration of heterodimeric B-RCAL. Mean values and SDs of three independent measurements are shown. **b** Binding of peptide 3 cyclized at only one cysteine pair or linear peptide 3. Cysteines in non-modified positions were substituted for serine. Additional singly bridged peptides are shown in Supplementary Fig. 8. Mean values and SDs of three (isomer 1a and 1b) or five (linear peptide) independent measurements are shown. **c, d** Alanine scan of linear peptide 3. **c** Sequences and $K_d$ measured by FP. Non-binding alanine mutants are highlighted in red. The values indicated are based on one measurement. **d** Binding response of peptides to calprotectin immobilized on an SPR chip. Peptides were injected at 500 nM for a single measurement. Non-binding alanine mutants are highlighted in red.

either to complete loss in binding (Phe8, Met10, Phe11, His14, and Tyr16) or had only little effect on the binding affinity (all other positions; $\Delta K_d < 2$-fold), as found via FP (Fig. 2c and Supplementary Fig. 5) and SPR (Fig. 2d). This result was in line with residues that were proposed to be most important based on the consensus sequences (Phe8, Pro9, Met10, and Phe11).

To identify the binding site, we co-crystallized linear peptide 3 with the recombinant calprotectin B-RCAL lacking the His$_6$-tag and with the heterodimer linker cleaved by 3C protease. The structure of the calprotectin tetramer with two bound peptide 3 molecules could be solved at 1.85 Å resolution (PDB entry 7QUV; Fig. 3 and Supplementary Table 2) by molecular replacement using a previously solved structure of calprotectin[25]. One copy of calprotectin (S100A8 and S100A9) and one copy of peptide constitute the asymmetric unit, while the calprotectin tetramer is formed by symmetry. The X-ray structure showed that peptide 3 binds in an elongated conformation, covering a large surface of 951 Å².

The N-terminal eleven amino acids of the peptide **RSPES-VAFPMF**QSHWYSG (bold; key amino acids found in Ala-scan are underlined) bound to the first heterodimer (S100A8[1], S100A9[1]), running along the seam formed by S100A8[1] and S100A9[1], and the C-terminal amino acids bound to a groove formed in a region where S100A9[1] meets S100A8 of the second heterodimer (S100A8[2]) (Fig. 3b). The binding region running over both heterodimers explains why peptide 3 binds to the tetrameric form but not to the dimer. The peptide forms seven H bonds and one charged interaction with calprotectin (Fig. 3c and Supplementary Table 3). Several amino acid side

chains, including those of the key residues Phe8, Met10, and Phe11, bind to small cavities to form hydrophobic contacts. The electron density map of the peptide shows that the middle and C-terminal regions of the peptide are well defined in the X-ray structure (Fig. 3d). An omit map in the absence of the modeled peptide unambiguously indicated that the positive features of the electron density came from the peptide (Supplementary Fig. 9). The binding mode of the linear peptide explained why isomer 1 of peptide 3 bound tightly to calprotectin but not isomers 2 and 3: Cys1 is in close proximity to Cys2, as well as Cys3 and Cys4 (Fig. 3e), and linking these pairs allowed the peptide to still bind in the elongated conformation. Bridging other pairs of cysteines would prevent the formation of many molecular contacts seen for the linear peptide in the structure.

**Peptide detects low nM calprotectin concentrations**

Having a peptide that binds to calprotectin through defined molecular contacts and with high affinity, we assessed whether it could detect calprotectin in sandwich-based diagnostic assays. We synthesized linear peptide 3 with biotin linked via a spacer (GSGSG) to either the N- or C-terminus (Fig. 4a and Supplementary Fig. 4) and tested these peptides in an ELISA-type assay (Fig. 4b and Supplementary Fig. 10a). Both peptides, pre-mixed with streptavidin-HRP, bound to calprotectin that was immobilized by an antibody in an ELISA plate (Fig. 4c). The signal detected with an HRP substrate correlated with the amount of immobilized calprotectin. The smallest concentration of calprotectin used was 1.6 nM (38 ng/mL), which is around 250-fold lower than the lower end of calprotectin found in the stool samples of healthy

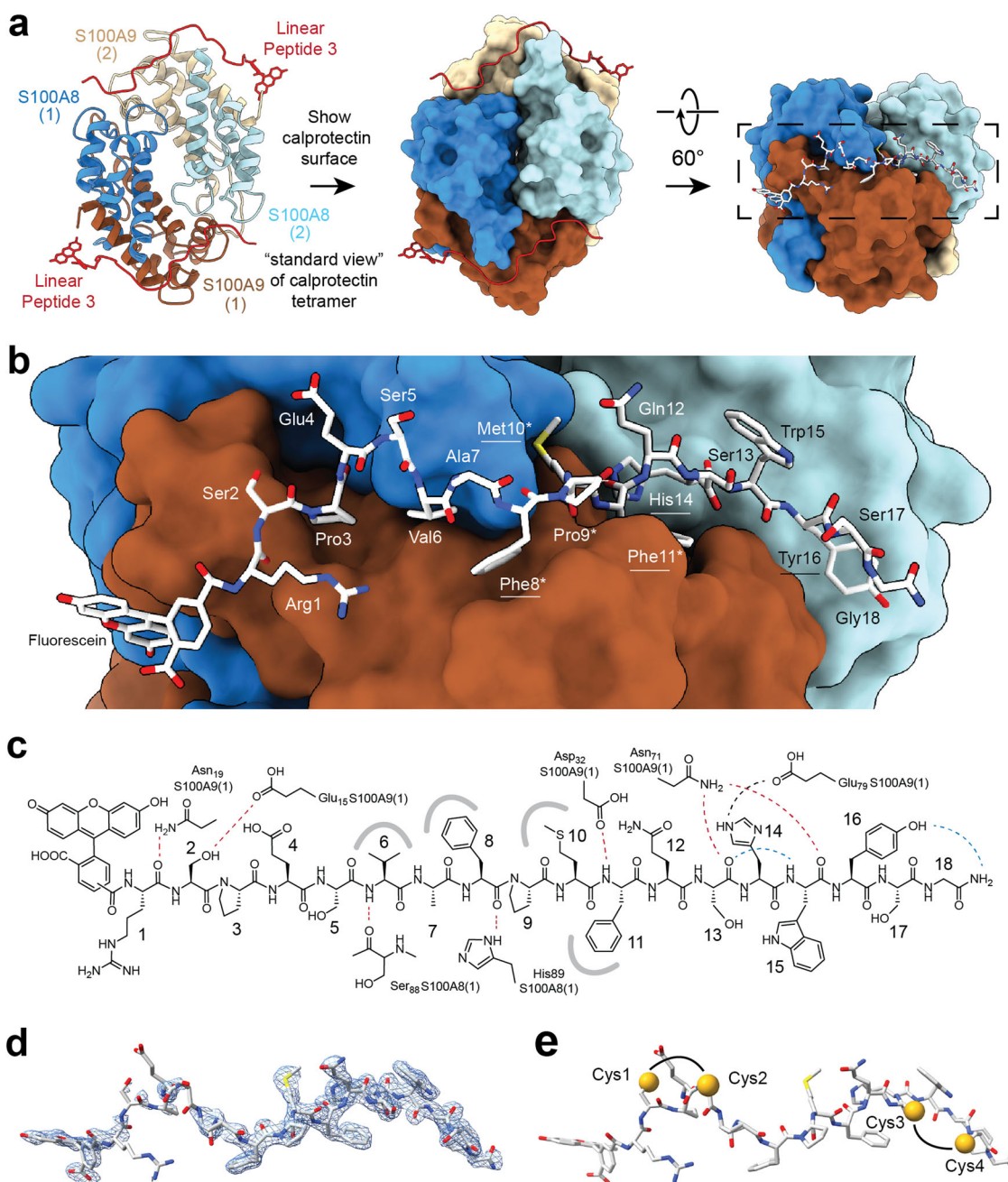

**Fig. 3 | X-ray structure of linear peptide 3 bound to tetrameric calprotectin.**
**a** Overview of X-ray structure. The first heterodimer is shown in the dark blue (S100A8[1]) and dark brown (S100A9 [1]), and the second one in light blue (S100A8[2]) and light brown (S100A9 [2]). Linear peptide 3 labeled at the N-terminus with fluorescein is shown in red. **b** Zoom-in representation of section indicated by a dashed line in (**a**). Amino acids of the consensus sequence are indicated by an asterisk. Key amino acids identified by the alanine scan are underlined. **c** Chemical structure of linear peptide 3 indicating inter- (red) and intramolecular (blue) H bonds and a charged interaction (black). **d** 2Fo-Fc electron density map of linear peptide 3 contoured at a 1.5σ level. **e** Linear peptide 3 indicates the amino acids that were cysteines in the double-bridged peptide isomer 1 of peptide 3.

persons (10 µg/g) and around tenfold lower than the mean found in the serum of healthy persons (around 0.5 µg/mL). At all the concentrations, including the lowest one, calprotectin was efficiently detected by the peptide, meaning that calprotectin can easily be detected even if the preparation of patient samples requires several-fold dilution. Linear peptide 3 biotinylated at the C-terminus (linear peptide 3-biotin) gave slightly stronger signals than the peptide immobilized via the N-terminus (biotin-linear peptide 3) for reasons we could not rationalize based on the X-ray structure and the good accessibility of both peptide ends. Given the better performance of the C-terminally conjugated peptide, we used the linear peptide 3-biotin for further

experiments. Calprotectin was also efficiently detected if linear peptide 3 was incubated for only 10 minutes (Supplementary Fig. 10b), suggesting that the peptide binding rate ($k_{on}$) should be suitable for rapid diagnostic assays where short incubation times are a prerequisite.

**Peptide is suited for lateral flow assays**
We next tested whether the peptide was suitable for detecting and quantifying calprotectin in LFAs. For a proof-of-concept, we immobilized linear peptide 3-biotin on streptavidin-coated gold nanoparticles (Strep-AuNP) and used half-strips (dip-sticks) with only the

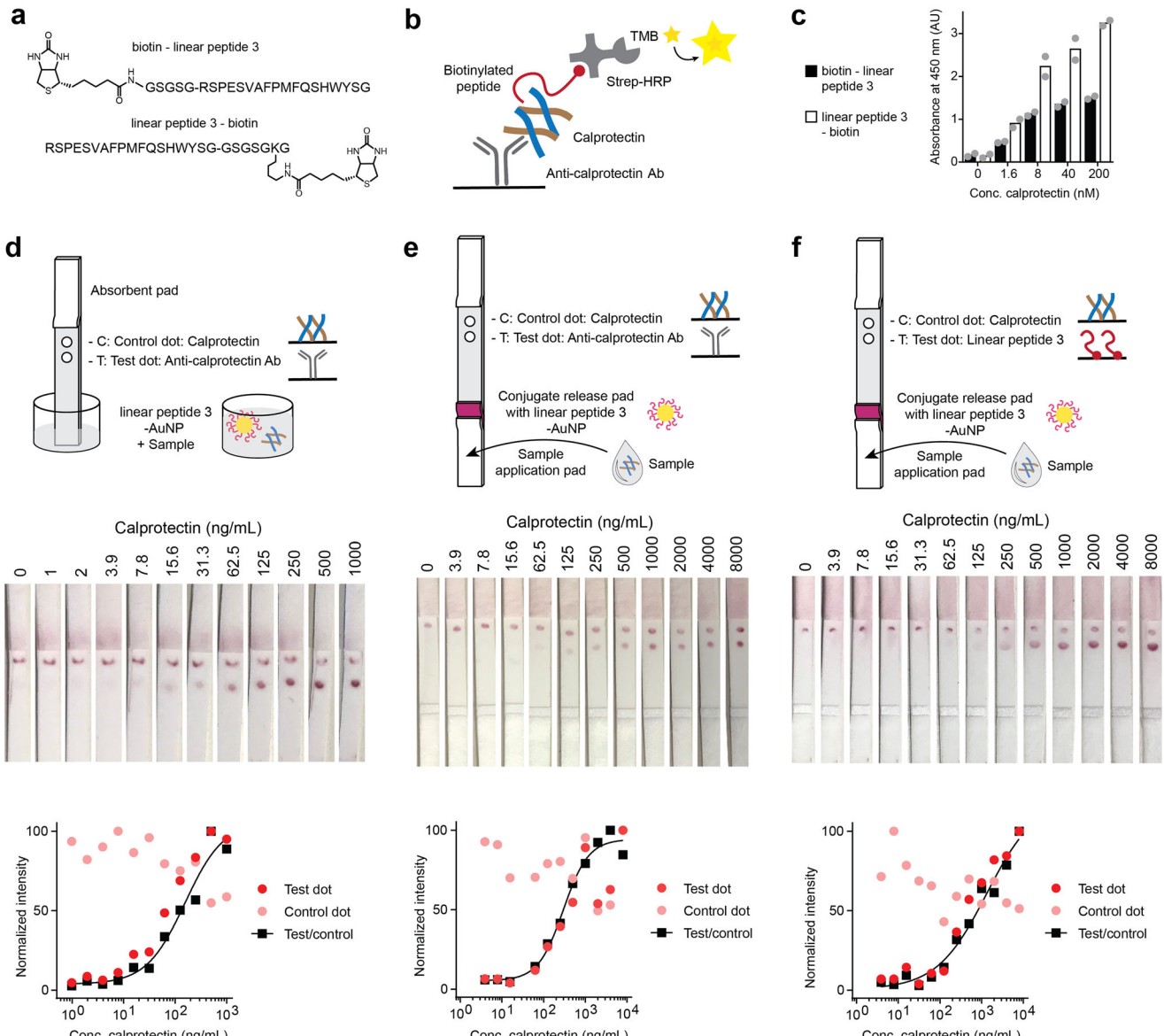

**Fig. 4 | Proof-of-concept detection of calprotectin by linear peptide 3.**
**a** Sequences and structures of biotinylated peptides. **b** Schematic representation of
ELISA format. **c** ELISA assay using linear peptide 3 biotinylated at the N- or
C-terminus. Different concentrations of recombinant calprotectin B-RCAL (indi-
cated as the concentration of the heterodimer) were captured on immobilized anti-
calprotectin antibody and detected with either of the two biotinylated peptides/
Strep-HRP. Average values (bars) and individual values (dots) of two measurements
are shown. **d** Schematic of a dipstick lateral flow assay. Detection of recombinant

calprotectin tetramer by dipstick lateral flow assay. Chase buffer was spiked with
the indicated concentrations of B-RCAL. **e**, **f** Full lateral flow assay where peptide
3-AuNP was deposited to the conjugate-release pad (red). Samples containing
calprotectin (B-RCAL) were added in 80 μL drops to the sample application pad.
The test dot (T) contains anti-calprotectin Ab (**e**) or neutravidin-bound biotinylated
linear peptide 3 (**f**). For panels **d**–**f**, the intensities of the T and C dots were nor-
malized to the T and C dots with the highest values, respectively. T/C ratios were
calculated and normalized to the highest ratio value and fitted to sigmoidal curves.

nitrocellulose membrane and the absorbent pad as shown in Fig. 4d.
The membrane was manually dotted in the corresponding regions
using anti-calprotectin rabbit polyclonal antibody (test) and calpro-
tectin (control). The optimal pH of the storage buffer for the peptide-
conjugated AuNP (linear peptide 3-AuNP) was identified by measuring
the UV-Vis profile in these conditions (Supplementary Fig. 11). The
stability was measured by incubating the peptide 3-AuNP beads for 18
days and then performing a UV-Vis analysis, which showed little var-
iation in signal, indicating low variability and high stability (Supple-
mentary Fig. 11). After dipping the half-strips into solutions with
various calprotectin concentrations and incubating for 15 min, the
beads were detected as red dots at the test and control positions,
and the intensities at the test zones correlated with the applied
concentrations of calprotectin, showing a low limit of detection of

15.6 ng/mL and a wide dynamic range (around 15.6 ng/mL to 1 μg/mL;
Fig. 4d and Supplementary Fig. 12).

With this proof-of-concept for the half-strip method, we per-
formed full LFAs. The stick contained a conjugate-release region to
which we deposited the linear peptide 3-AuNP beads and a sample
application pad to which we pipetted 80 μL drops of the sample con-
taining calprotectin (B-RCAL) (Fig. 4e). The calprotectin moved in the
chase buffer, associated with the deposited linear peptide 3-AuNP
beads, and the beads bound to anti-calprotectin mAb in the test region
and calprotectin in the control region of the full LFA. The concentration
range that could be reliably quantified was between 15.6 and 1000 ng/
mL of calprotectin, which is comparable to LFAs employing two anti-
bodies (e.g., 50–1000 ng/mL, sCAL test from BÜHLMANN Labora-
tories). The dynamic range is perfectly suited to detect calprotectin in

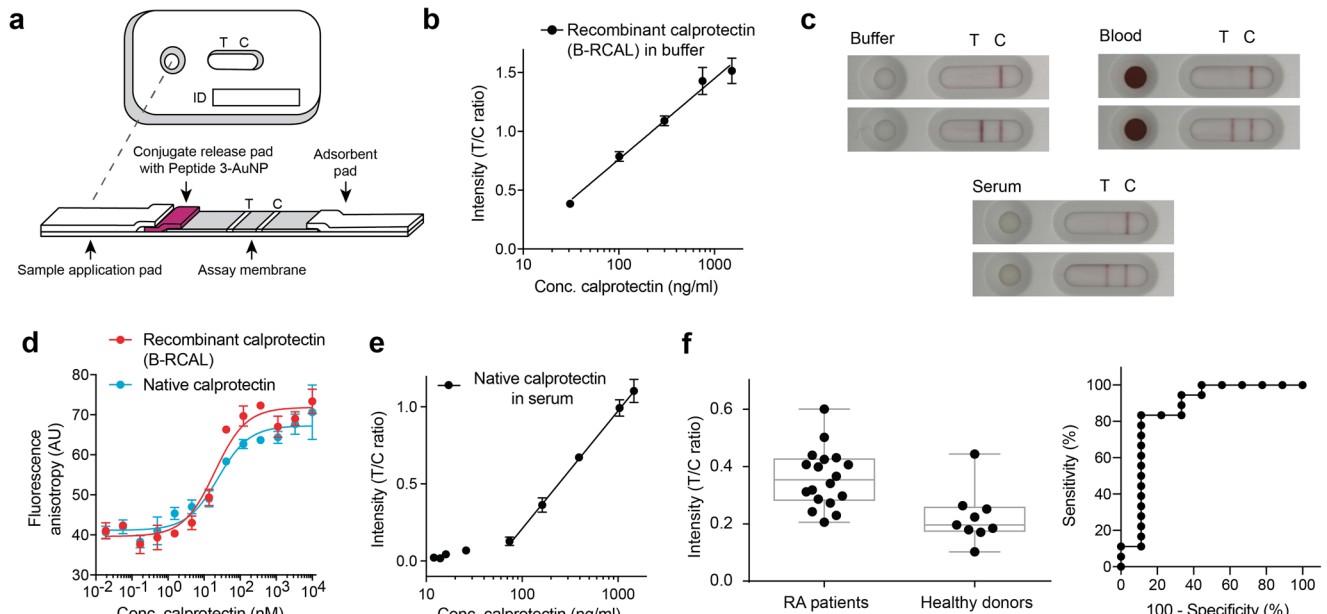

**Fig. 5 | Lateral flow assay with native calprotectin and patient samples.**
**a** Assembled lateral flow test cassettes. **b** Calibration curve of the lateral flow device using recombinant calprotectin (B-RCAL) showing the T/C ratio. Mean values and SDs of five replicates are indicated. **c** LFA applying recombinant calprotectin (B-RCAL) in buffer, blood plasma, or blood. Photographs of the LF strips after 15 min incubation without calprotectin (upper strips) or recombinant calprotectin (B-RCAL; 80 µL droplet of 950 ng/mL) (lower strips). Video recordings of the assays are provided as Supplementary Material. **d** Comparing the binding of fluorescein-labeled linear peptide 3 to recombinant calprotectin and native calprotectin

(obtained from granulocytes). Mean values and SDs of three replicates are indicated. **e** Calibration curve of the lateral flow device using native calprotectin in human serum, showing the T/C ratio. Mean values and SDs of five replicates are indicated. **f** Analysis of serum samples from 18 rheumatoid arthritis patients and nine healthy donors. The left graph shows the averages of duplicate measurements of the samples. Box plots of the median (the line between two boxes) with interquartile range (boxes) and the minimum to maximum range (whiskers) are shown. The right panel provides a ROC curve analysis with the area under the curve (AUC) equal to 0.86.

the clinically relevant range from 1–15 µg/mL after diluting samples around 20-fold in assay chase buffer. The large dilution factor allowed by the sensitive detection is ideal for efficiently adjusting solvent, pH, and ion concentration to an optimal LFA running buffer.

We finally tested if bead-bound calprotectin could be captured in the test region by linear peptide 3 instead of antibody, thus forming an LFA completely independent of antibodies (Fig. 4f). We immobilized the biotinylated linear peptide 3 in the test region as a complex with neutravidin. The LFA performed well, with the beads forming clear dots at the test region with intensities correlating with the applied calprotectin concentrations. The result showed that some binding sites of the bead-bound calprotectin tetramer were available to bind additional peptide 3 in the test region. The sensitivity and dynamic range of the antibody-free LFA are perfectly suited for the diagnostic detection of calprotectin.

**Diagnostic detection of native calprotectin**
We next developed LFA cassettes using conjugate-release pads coated with linear peptide 3-AuNP beads, with the test and control reagents blotted as lines rather than dots on the nitrocellulose strips (Fig. 5a). The LFAs in cassettes also performed well, showing a linear correlation between signal and calprotectin (B-RCAL) concentration in a range from 30–2000 ng/mL (Fig. 5b). Tests performed with the cassettes and calprotectin in complex matrices such as plasma and blood (spiked with 950 ng/mL and no calprotectin in the negative control) also showed clear results, indicating that the assay format is suitable for analyzing patient samples derived from blood and plasma (Fig. 5c).

As native calprotectin is slightly different from the recombinant protein (B-RCAL) used in the above experiments, we next tested the binding of linear peptide 3 to native calprotectin derived from granulocytes. The peptide bound to native calprotectin with essentially the same dissociation constant as assessed by FP for B-RCAL ($K_d = 20 \pm 4$ nM for B-RCAL and $26 \pm 13$ nM for native calprotectin;

Fig. 5d), meaning that linear peptide 3 is suitable for the detection of calprotectin in patients. As found for the recombinant protein, the linear peptide 3 bound to the tetrameric protein but not to the dimer (obtained by adding 1 mM EDTA; Supplementary Fig. 13). As expected, the peptide could not bind to a non-tetrameric mutant (E78A) of the recombinant calprotectin (Supplementary Fig. 13).

We finally applied the LFA cassettes for quantifying calprotectin in patient samples, focusing on serum rather than feces samples due to their availability and the interest in assessing the compatibility of the peptide-based LFA with serum components. We used serum samples of nine healthy donors and 18 rheumatoid arthritis patients, the latter of which were previously shown to have slightly elevated levels of calprotectin[26]. We first tested the LFA devices using human serum samples containing different amounts of native calprotectin (120 to 14,700 ng/mL) that were diluted tenfold with chase buffer. As these samples showed a good correlation between analyte concentration and signal (Fig. 5e), we also diluted the patient serum samples tenfold in chase buffer. The calprotectin concentrations could be efficiently determined for all patient samples and showed elevated levels for the RA patient group (Fig. 5f). Receiver operating characteristic (ROC) curve analysis revealed an area under the curve (AUC) of 0.86 (Fig. 5f), indicating good discrimination between healthy donors and rheumatoid arthritis patients. The clear signal lines obtained for samples derived from many different patients/serums indicated a robust performance of the peptide-based LFA.

While we did not yet assess the stability and storage requirements for the peptide 3-AuNP-based LFA in a systematic study, we found that cassettes that were stored for 11 months in a dry atmosphere detected the same concentrations of calprotectin, hinting good stability (Supplementary Fig. 14). A potentially vulnerable site of peptide 3 is the methionine residue that can oxidize, as seen for some of the peptide variants synthesized (Supplementary Fig. 4; small peak eluting around one minute before the peptide). We tested several mutants of peptide

3 and found that substitution of the Met10 to leucine largely conserves the binding affinity ($K_d = 30.6 \pm 4.6$ nM; Supplementary Fig. 5). In contrast to peptide 3 in which a small quantity was oxidized upon storage in solution, the Met10Leu mutant did not show any oxidation at all, suggesting that a calprotectin LFA may be developed that can be stored for a very long time.

## Discussion

To address the need for stable and accurate non-invasive tests for calprotectin, we identified high-affinity peptide binders to this important inflammation marker. As assessed by an alanine scan and X-ray structure analysis, the identified peptide engages with calprotectin through a large number of well-defined contacts, such as several H-bonds, an electrostatic interaction, and several contacts of hydrophobic side chains with small hydrophobic cavities on the surface of the target protein. The best peptide that we isolated has a fast binding rate and a relatively slow off-rate, making it an ideal recognition element for diagnostic assays.

We were able to find the peptide by screening the largest ever reported peptide phage display library. While the binding affinity is likely sufficient for immediate application, it is tempting to speculate that it may be improved by several magnitudes by further engineering the peptide, such as by replacing some of the amino acids with non-canonical ones that confer increased affinity, a general strategy for engineering therapeutic peptides. Though this is not a priority for the calprotectin assays due to the more than adequate coverage of the relevant concentration range, it could be an option for developing assays for different targets present in reduced quantities.

In applying these peptides for detecting and quantifying calprotectin in an ELISA assay and by LFAs, we assembled robust and sensitive assays that detected calprotectin concentrations much lower than those in patient samples, meaning that samples can be diluted by large factors, if required. The robustness, high reproducibility, and ease of developing the assays observed herein can be attributed to a large extent to the ability of synthesizing the peptides in a homogenous, pure form and as conjugates for site-directed and convenient immobilization. Given their synthetic nature, peptides can be obtained at higher homogeneity and purity than proteins overall (>95% purity in this study), and we expect that LFAs based on peptides will show a lower batch-to-batch variation. Extensive studies will be required for side-by-side comparing the peptide-based LFA to existing antibody-based LFAs.

Taking advantage of the two binding sites of the calprotectin tetramer, we developed an antibody-free LFA, as the peptide could serve as both the calprotectin capture and detection agent. This lack of antibodies is expected to give peptide-based LFAs more flexible storage requirements (e.g., room temperature) and longer stability. This is mainly because, unlike proteins, peptides are not folded and cannot unfold, making them highly heat-stable. In this regard, the experiments performed in this study are limited by their immobilization as biotin conjugates to streptavidin/neutravidin, which are proteins and may have similar liabilities as antibodies. Regardless, peptides are regularly immobilized on materials such as gold via cysteines (stable gold–sulfur interaction) or can be covalently bound to polymers, which provide an efficient and simple next step for further removing proteins from these assays.

The synthetic peptide-based ligands of calprotectin offer a basis for the development of robust, antibody-free diagnostic assays with smaller error margins and less restrictive storage requirements. As the recognized epitope on calprotectin is known in molecular detail and the peptide measures specifically the tetrameric state of calprotectin, assays based on it exclude an important source of inter-assay variability found for current calprotectin tests. Though the presented peptides provide an immediate solution for accurately detecting calprotectin, future work will remove all proteins from the assay and will characterize the longer-term stability and storage of subsequent LFAs. We expect that these peptides will provide a simple, accurate, and non-invasive test for quantifying calprotectin in fecal or blood samples, for diagnosing and monitoring IBDs and other diseases. In addition, the successful quantification of calprotectin with a phage display-selected peptide may stimulate the broader application of peptide-based affinity reagents for diagnostic assays.

## Methods

### Ethical aspects

Research complies with all relevant ethical regulations. Human serum samples used in this work were purchased by BÜHLMANN Laboratories from In.Vent Diagnostica GmbH (Henningsdorf, Germany) who had received ethical approval from the Freiburg Ethics Committee International (feki). In.Vent Diagnostica GmbH had received consent from patients to use the samples for research.

### Biotinylation and immobilization of calprotectin on magnetic beads

Recombinantly expressed fused calprotectin $His_6$-linker-S100A9-linker-S100A8 (B-RCAL) in 20 mM HEPES, 100 mM NaCl, pH 7.5, 2.5% (v/v) glycerol, 1 mM DTT buffer was provided by BÜHLMANN Laboratories AG Schönenbuch, Switzerland. Calprotectin was immobilized on magnetic beads by random biotinylation of amino groups and in addition to streptavidin- or neutravidin-coated beads. The two types of beads were used alternatively to disfavor the enrichment of streptavidin- or neutravidin-specific peptides. Neutravidin beads were prepared by reacting 6 mg of neutravidin (Pierce) with 10 mL of tosyl-activated magnetic beads (Dynal M-280, Invitrogen) according to the supplier's instructions.

Calprotectin was biotinylated by incubating 500 µL of B-RCAL (10 µM) with 5 µL of EZ-Link™ Sulfo-NHS-LC-Biotin (Thermo Fisher Scientific) (10 mM; final conc. 200 µM, 20-fold molar excess) in 20 mM HEPES (pH 7.5), 150 mM NaCl, and 2 mM $CaCl_2$. The reaction was incubated for 1 h at room temperature. The protein was separated from the unreacted reagent using a PD-10 column (GE Healthcare) and buffer containing 20 mM HEPES, 150 mM NaCl, pH 7.5, 2 mM $CaCl_2$, and 1 mM TCEP. The biotinylated protein was concentrated and stored at −80 °C. To assess the efficiency of biotinylated calprotectin, 2 µg of biotinylated protein were captured on 30 µL of magnetic streptavidin beads, and the immobilized protein was analyzed by SDS-PAGE as described in ref. 27.

Biotinylated protein was immobilized on magnetic beads by incubation of protein and pre-washed beads in 200 µL washing buffer (10 mM Tris, pH 7.4, 150 mM NaCl, 10 mM $MgCl_2$, and 2 mM $CaCl_2$) for 30 min at room temperature on a rotating wheel (10 rpm). Then, 3 µL of biotin (1 mM) was added to block the remaining biotin-binding sites on the beads for another 30 min. The beads were washed three times with 1 mL of washing buffer and resuspended in 400 µL washing buffer with 1% BSA and 0.1% Tween-20.

### Amino acid sequence of B-RCAL

MMHHHHHHLEVLFQGPMTCKMSQLERNIETIINTFHQYSVKLGHPDTLN
QGEFKELVRKDLQNFLKKENKNEKVIEHIMEDLDTNADKQLSFEEFIMLMA
RLTWASHEKMHEGDEGPGHHHKPGLGEGTPLEVLFQGPMLTELEKALNS
IIDVYHKYSLIKGNFHAVYRDDLKKLLETECPQYIRKKGADVWFKELDINTD
GAVNFQEFLILVIKMGVAAHKKSHEESHKE

Number of amino acids: 231
Molecular weight: 26911.80
Theoretical pI: 6.08

### Cloning of monocyclic peptide library (Library 3)

The library was constructed using a whole-vector PCR strategy as described in ref. 28, using the 24 DNA primers (5′→3′) shown in Supplementary Table 1. The primers code for peptides of the format

(X)$_{0-3}$C(X)$_{3-8}$CX (X = any amino acid; C = cysteine) with a length of 6 to 14 amino acids, and a Gly–Ser–Gly linker connecting the peptides to the phage pIII protein. They contain a *Sfi*I cleavage site for the circularization of the PCR products. The degenerated primers were synthesized using NNK codons in the random amino acid positions (Macrogen). For the whole-vector PCR reaction, the vector fd-tet-PK15[20] was used as a template, and the DNA 5′-GCTTCATCTGCCGGA-CAATTACG-3′ as reverse primer. The template vector is based on the vector fd-tet[29] and contains the 17-amino acid sequence PK15[30] and a leader sequence, as shown in the following. 5′-GTGAAAAAATTATT ATTCGCAATTCCTTTAGTTGTTCCTTTCTATGCGGCCCCAGCCGGCCAT GGCAGCATGTAGCGATCGTTTTCGTAATTGTCCGGCAGATGAAGCAC TGTGTGGTGGTTCTGGCGCTGAAACTGTTGAAAGTTGT-3′

The PCR reactions with the 24 primers were performed in separate reactions. The total reaction volume was 50 μL and contained one of the forward primers (10 μM), the reverse primer (10 μM), dNTP mix (10 mM each), 40 ng of fd-tet PK15 template, 10 μL of 5× Phusion buffer (Thermo Fisher Scientific) and 4 units of Phusion polymerase (Thermo Fisher Scientific). The following PCR program was used: 2 min at 95 °C, 25 cycles of 30 s at 95 °C, 45 s at 55 °C, and 7 min at 72 °C, and final elongation for 7 min at 72 °C. The PCR products were purified by electrophoresis on a 1% agarose gel in TAE buffer containing 1 mM guanosine, gel extracted using a kit (QIAquick gel extraction kit, Qiagen), and the products were pooled.

The two ends of the whole-plasmid PCR product were digested with *Sfi*I (CAT# R0123S, New England Biolabs) as follows. In a 0.5-mL tube, 10 μg of the PCR product and 20 μL of 10× Cutsmart® buffer were pipetted and diluted by filling up to 198 μL with ddH$_2$O. *Sfi*I (2 μL, 40 units) was added, the reaction was mixed well by pipetting and was incubated for 4 h at 50 °C. The digested PCR product was purified with a DNA purification kit (QIAquick® Gel Extraction Kit, Qiagen). The DNA was eluted from the columns (2 columns for 10 μg of DNA) with 10 mM Tris·HCl buffer, pH 8.5 (three times 20 μL).

The DNA was self-ligated in a reaction with 50 μL 10× T4 DNA ligase buffer, 50 Weiss units of T4 ligase (Thermo Fisher Scientific), and ddH$_2$O up to a final volume of 500 μL. The reaction was incubated at 20 °C for 4 h. Ligase inactivation was performed by incubating the tube at 70 °C for 10 min. Subsequently, ligated DNA was purified using a commercial kit (QIAquick gel extraction kit, Qiagen) and eluted from the columns using ddH$_2$O. DNA was electroporated into commercial *E. coli* TG1 cells (Lucigen). About 25 μL of cells were electroporated with 1 μg of DNA. After electroporation, 1 mL of pre-warmed commercial recovery medium (Lucigen) were added to the cells and finally incubated at 37 °C for 1 h with shaking (200 rpm). Cells were plated on nine large (20 cm diameter) 2YT/ tetracycline (100 μg/mL) plates. Plates were incubated at 37 °C overnight. The library size was determined by plating cell dilutions on 2×YT/tetra-cycline agar plates. Electroporation of the DNA into TG1 *E. coli* cells yielded $2.6 \times 10^8$ colonies. Colonies were collected from the plates with 2YT medium/20% v/v glycerol and stored at −80 °C. Ninety-six library clones were sequenced by Sanger sequencing (Macrogen) to evaluate the library quality.

### Production of phage peptide Libraries 1 and 3
Phage for Libraries 1 and 3 were produced as described in Kong, X.D. et al. (Supporting Information, sub-chapter "Phage selection against streptavidin")[28] with a few modifications. Six library glycerol stocks (Library 1) or one glycerol library stock (Library 3) were added each to a 0.5 L 2YT/tetracycline (100 μg/mL) culture for inoculation. Before incubation, a sample was taken to determine the number of viable cells and to ensure that this number matches or exceeds the library size. The culture was grown at 30 °C overnight with shaking (200 rpm). On the next day, the cells were pelleted at $5500 \times g$ (5000 rpm in an F8-6x 1000 y rotor and a Sorvall Bios centrifuge) at 4 °C and the supernatant containing the phage was transferred to

new centrifugation tubes. Phage precipitation was performed by adding 125 mL of cooled PEG/NaCl solution (20% PEG-6000 (w/v), 2.5 M NaCl), followed by incubation for 30 min on ice. Phages were then centrifuged at $9300 \times g$ (6500 rpm in an F8-6x 1000 y rotor and a Sorvall Bios centrifuge) for 45 min at 4 °C. The supernatant was carefully discarded and the phage pellets were resuspended in 15 mL degassed reaction buffer (20 mM NH$_4$HCO$_3$, pH 8.0, and 5 mM EDTA). The remaining cells were removed by centrifugation at $5500 \times g$ for 15 min at 4 °C. Aliquots of phage were taken before and after the precipitation to determine the phage titers.

The cysteine residues of the peptides were reduced by adding 1 mM TCEP for 30 min at 25 °C. The reducing agent was subsequently removed by phage precipitation, adding PEG/NaCl, centrifugation, and phage resuspension in 18 mL of the degassed reaction buffer. For each cyclization reaction with a different linker, 4.5 mL of phage were prepared in a separate tube and 0.5 mL ACN containing bis-electrophile reagent were added, to reach final concentrations of 30 μM (linker 1 and 4) or 40 μM (linker 2, 3, 5, and 6). The reactions were incubated at 30 °C for 1 h and the excess of cyclization reagent was removed by phage precipitation using PEG/NaCl. The phage pellets were resuspended in 5 mL binding buffer (10 mM Tris, pH 7.4, 150 mM NaCl, 10 mM MgCl$_2$, 2 mM CaCl$_2$, 1 % BSA, and 0.1% Tween-20) and stored at 4 °C overnight.

### Production of phage peptide Library 2
Phage for Library 2 were produced as described in ref. 18 (chapter "Phage production") with a few modifications. One library glycerol stock was inoculated in 1 L 2YT/ampicillin (100 μg/mL) culture with 100 mM glucose to reach an initial OD$_{600}$ = 0.1. The culture was grown at 37 °C until it reached an OD$_{600}$ of 0.5. The culture was then infected with hyperphage M13 K07ΔpIII (Progen Biotechnik GmbH) at a multiplicity of infection (MOI) of 10 and incubated for 15 min without shaking, and 45 min at 200 rpm at 37 °C. Cells were pelleted and resuspended in 1 L of 2YT/ampicillin (100 μg/mL) + kanamycin (50 μg/mL) medium, and incubated at 30 °C overnight with shaking at 250 rpm. The next steps of phage precipitation, cysteine reduction and modification, panning, and sequencing were performed in the same way as described for Libraries 1 and 3.

### Phage display selections
Biotinylated recombinant calprotectin (B-RCAL) (1, 0.5, and 0.05 μg for Library 1 and rounds 1–3; 5, 2.5, 0.2, 0.02, and 0.002 μg for Library 2 and rounds 1–5; 5, 2.5, and 1 μg for Library 3 and rounds 1–3) immobilized on magnetic beads (20 μL of streptavidin beads in the first, third, and fifth round, and 10 μL of neutravidin beads in the second and fourth round) was incubated with phage libraries for 30 min with rotation. Unbound phage was removed by washing the beads eight times with washing buffer containing 0.1% Tween-20 and three times with washing buffer. The beads were suspended in 100 μL glycine buffer (20 mM, pH 2.2) and incubated for 5 min to elute the phage. The solution was neutralized by adding 100 μL of Tris-Cl buffer (1 M, pH 8.0).

The eluted phage were added to 10 mL of exponentially growing *E. coli* TG1 cells at OD$_{600}$ = 0.4. After incubation at 37 °C for 30 min without shaking, the freshly infected bacteria were plated on selective 2YT plates (100 μg/mL tetracycline for Library 1 and 3; 100 μg/mL ampicillin for Library 2) and grown overnight at 30 °C. Aliquots of phage were taken before and after panning to determine the phage titers. Bacterial cells of the colonies grown overnight were recovered in a 2YT medium containing 20% glycerol, flash-frozen, and stored at −80 °C until the next round of selection.

For the next rounds of selection, the scale for phage production was reduced to 25 mL (one culture for each cyclization reagent), and the rest of the solution volumes were adjusted accordingly. Enriched phage peptides were sequenced by Sanger sequencing (Macrogen)

and the sequences were grouped based on similarity into consensus sequences.

## Sequencing of phage peptides

Individual colonies were picked to inoculate 100 µL 2YT cultures in 96-well plates. Cultures were grown for 2 h at 37 °C and 2 µL of the cultures were used for colony PCR. The PCR reactions with volumes of 30 µL were prepared by distributing 28 µL of a master mix containing the following reagents and adding 2 µL of the cell cultures. Master mix: Water, Taq buffer, forward primer (20 nM each, final concentration), reverse primer (20 nM each, final concentration), dNTP mix (200 µM each, final concentration), and one unit Taq polymerase (New England Biolabs) for each PCR reaction. The PCR mixture was immediately subjected to thermocycling using the following program: initial denaturation for 30 s at 95 °C, followed by 30 cycles of 30 s at 95 °C, 30 s at 50 °C and 1 min at 68 °C, and final elongation for 5 min at 68 °C. Volumes of 3 µL were analyzed on a 1% agarose gel (containing 0.005% [v/v] ethidium bromide) for the presence of a band at around 1000 base pairs. The PCR plates containing the PCR reactions, covered with cap-strips, were sent for sequencing with the primer CACCTC-GAAAGCAAGCTGATAAACC (Library 1 and 3) and the primer GTGT GGAATTGTGAGCGGATAAC (Library 2) (Macrogen).

## Solid-phase synthesis of peptides

Peptides were synthesized by solid-phase peptide synthesis (SPPS) using standard Fmoc-chemistry and DMF as a solvent on a MultiPep RSi parallel peptide synthesizer (Intavis). The peptides were typically prepared at a 25 µmol scale in disposable 5 mL reactors using rink amide AM resin as a solid phase. The resin was swollen by repetitive addition and removal of DMF. The amino acids were coupled twice (2.15 equiv.; 0.18 M final conc.) at room temperature for 45 min using HATU (2.05 equiv.; 0.17 M final conc.) as coupling reagent and 4-methylmorpholine (NMM; 4.7 equiv.; 0.4 M final conc.) as a base. After the coupling reaction, two washing cycles with DMF (1.8 mL) were performed. N-terminal amines remaining free after coupling were capped using acetic anhydride (5% v/v) and lutidine (6% v/v) at RT for 30 min (0.4 mL). Seven washing cycles were performed. Fmoc groups were deprotected twice using piperidine (20% v/v) in DMF (0.45 mL) at room temperature for 5 min, followed by seven washing cycles (1.8 mL). The N-terminal carboxyfluorescein moiety was incorporated manually by reacting twice with 5(6)-carboxyfluorescein (3 equiv.), HATU (4 equiv.), and DIPEA (3 equiv.) in 2 mL DMF for 45 min. The resin was washed three times with DMF (5 mL), treated three times with piperidine (20% [v/v] in DMF, 5 mL), washed three times with DMF (5 mL), and three times with DCM (5 mL).

Total cleavage of peptides was performed by adding 5 mL of a standard cleavage cocktail (90% TFA, 2.5% thioanisol, 2.5% H₂O, 2.5% 1.2-ethanedithiol, 2.5% phenol) and incubation for 4 h while shaking. The peptide-containing solution was collected by vacuum filtration and the peptides ether-precipitated as follows. A volume of 50 mL of ice-cold diethyl ether were added to the peptides, incubated for 30 min at −20 °C and then centrifuged at 2700 × g for 10 min (3600 rpm in a 19/w-9 rotor and a Sigma 4-16KS centrifuge) at 4 °C. Peptide pellets were washed with 35 mL of diethyl ether and centrifuged again to remove remaining diethyl ether. Crude peptides were cyclized directly with bis-electrophile linkers and purified by HPLC.

## Cyclization of peptides

Around 20 to 25 mg crude or HPLC-purified peptide was dissolved in 30% (v/v) ACN and 70% (v/v) aqueous buffer (60 mM NH₄HCO₃, pH 8.0) to reach a concentration of 1 mM (typically around 10 mL) and the cyclization reagent was added in ACN (3 equiv., 100 µL). The reaction mixture was incubated at 30 °C for 1 h, and the completion of the reaction was assessed by LC-MS. The reaction was stopped by the

addition of concentrated HCOOH (200 µL) and the cyclized peptide was lyophilized.

## HPLC purification of peptides

Quantities of 20 to 25 mg peptide were purified by HPLC (Prep LC 2535 HPLC, Waters) using a preparative C18 reversed-phase column (SunfireTM prep C18 OBD 10 µm, 100 Å, 19 × 250 mm, Waters) and applying a flow rate of 20 mL/min and an appropriate linear gradient over 40 min (A: 99.9% H₂O and 0.1% TFA; B: 99.9% ACN and 0.1% TFA). Absorbance was monitored at 220 nm. Fractions containing the desired peptide were pooled together and lyophilized.

Quantities of 5 to 10 mg peptide were purified by HPLC (Prep LC 2535 HPLC, Waters) using a semi-preparative reversed-phase C18 column (X-bridge peptide BEH C18 5 µm, 300 Å, 10 × 250 mm, Waters) applying a flow rate of 6 mL/min and an appropriate linear gradient in 40 min (A: 99.9% H₂O and 0.1% TFA; B: 99.9% ACN and 0.1% TFA). Absorbance was monitored at 220 nm. Fractions containing the desired peptide were pooled together and lyophilized.

## Analytical HPLC and MS analysis

The purity of the peptides was assessed by analysing around 20 µg of the peptide by RP-HPLC (1260 HPLC system, Agilent) using a C18 column (ZORBAX 300SB-C18, 5 µm, 300 Å, 4.6 × 250 mm, Agilent). Peptides were run at a flow rate of 1 mL min⁻¹ with a linear gradient of 0–100% of solvent B over 15 min (A: 94.9% H₂O, 5% ACN, and 0.1% TFA; B: 99.9% ACN and 0.1% TFA). The mass was determined by electrospray ionization mass spectrometry (ESI-MS) in positive ion mode on a single quadrupole liquid chromatography-mass spectrometer (LC-MS-2020, Shimadzu) and the data were analyzed using the Shimadzu LabSolutions software.

## Chemical synthesis of double-bridged peptide isomers

Peptides with defined pairs of cysteines bridged by chemical linkers were synthesized in a similar manner as described in ref. 19 (Supporting Information, sub-chapter "Peptide cyclization by bridging specific pairs of cysteines"), with small changes as follows. The peptides were synthesized with two cysteines protected with diphenylmethyl (Dpm) groups and two with p-methoxytrityl (Mmt), instead of the previously used trityl (Trt) group. After the linear synthesis of the peptide and before the total cleavage, the resin was treated with 5 mL of TFA:TIS:DCM (1:5:94) in the fritted syringe for 8 × 2 min. The resin was washed three times with DCM and three times with DMF. About 1.5 equiv. of cyclization reagent and 4 equiv. of DIPEA in 4 mL of DMF were added and the reaction mixture was shaken for 1 h at room temperature. The reaction solution was removed and the resin was washed three times with DCM. Then, the resin was subjected to global deprotection with 90% TFA, 2.5% thioanisol, 2.5% H₂O, 2.5% 1.2-ethanedithiol, 2.5% phenol, for 6–8 h to assure that all the Dpm groups were removed. Ether purification and HPLC purification were performed as described above. The second linker was introduced, adding 1.5 equiv. of cyclization reagent. The peptide was purified by HPLC.

## Chemical synthesis of biotinylated peptides

Linear peptide 3 biotinylated at the N-terminus (biotin-linear peptide 3) was manually synthesized by conjugating biotin to the N-terminus of peptide 3 that was still on the solid phase, as follows. Biotin (4 equiv.), HATU (4 equiv.), and DIPEA (3 equiv.) 2 mL DMF were added to the resin for 45 min. The resin was washed with DMF (3 × 5 mL) and DCM (3 × 5 mL).

Linear peptide 3 biotinylated at the C-terminus (linear peptide 3-biotin) was synthesized by conjugating biotin to a lysine residue near the C-terminus as follows. Peptide 3 was synthesized using Fmoc-Lys(Dde)-OH as the second amino acid, and the last amino acid was incorporated as Boc-Arg(Pbf)-OH. After the completion of

the automated synthesis, the protecting group of Lys was manually removed with 2% hydrazine in DMF and biotin was manually incorporated at the deprotected amino group using the same reaction conditions as above. The peptide was fully deprotected and removed from the resin with the standard cleavage cocktail described above.

## Affinity measurement by fluorescence polarization

Calprotectin was serially diluted in 20 mM HEPES, 100 mM NaCl, 2 mM CaCl$_2$, pH 7.5, and 1 mM DTT with 0.01% (v/v) Tween-20. Volumes of 16 µL of protein were added to 4 µL of fluorescent peptide (20 nM final concentration) in black 384-well microtiter plates. Fluorescence anisotropy was measured on a microwell plate reader (Infinite M200Pro, Tecan) with filters for excitation at 485 nm and emission at 535 nm. The dissociation constant ($K_d$) was calculated using a following Eq. (1) using Prism 5 software (GraphPad):

$$A = A_f + \left(A_b - A_f\right) \times \left\{\frac{[L]_T + K_D + [P]_T - \sqrt{\left([L]_T + K_D + [P]_T\right)^2 - 4[L]_T[P]_T}}{2[L]_T}\right\}$$

(1)

where A is anisotropy, $A_f$ and $A_b$ are the anisotropy values for free and bound ligands, respectively. $[L]_T$ is the concentration of the total fluorescent ligand and $[P]_T$ the concentration of the protein.

For measuring binding to calprotectin dimer, the CaCl$_2$ in the dilution buffer was replaced with EDTA (1 mM).

## Affinity measurement by surface plasmon resonance

SPR measurements were performed using a Biacore™ 8 K instrument (GE Healthcare) using the Biacore™ 8 K Control software. Recombinant calprotectin (B-RCAL) was diluted with 10 mM acetate buffer (pH 5.0) to reach a concentration of 5 µg/mL and immobilized on a CM5 series S chip using the standard amine coupling method in running buffer (10 mM HEPES pH 7.4, 150 mM NaCl, 2 mM CaCl$_2$, and 0.005% v/v Tween-20) at 25 °C and the following procedure. At a flow of 10 µL/min, the following solutions were injected: 420 s 0.4 M EDC, 0.1 M NHS; 100 to 200 s protein; 500 s 1 M ethanolamine-HCl pH 8.5. Typical immobilization levels were 3000 resonance units (RUs). A reference cell was treated in the same way but without the injection of protein. A single concentration for each peptide (500 nM) was injected in a running buffer containing 0.5% (v/v) DMSO to measure the binding level. For the measurement of binding kinetics and dissociation constants, five serial dilutions (threefold) of peptides were prepared in running buffer containing 0.5% (v/v) DMSO and analyzed in single cycle kinetics mode with the contact and dissociation times of 90 s and 120 s, respectively. Data were analyzed using Biacore™ 8 K Evaluation software.

## Protein size determination by dynamic light scattering

Dynamic light scattering (DLS) experiments were performed using a Stunner instrument (Unchained Labs). Briefly, 2 µl of B-RCAL at 3.1 mg/ml were diluted with 4 µl of buffer (20 mM HEPES, 100 mM NaCl, pH 7.5 supplemented with either 10 mM EDTA, or 2 mM CaCl$_2$) and incubated for 1 h at room temperature. Next, 2 µl of the diluted sample were added to the Stunner 96-well plate and submitted to DLS measurement. Data were analysed using the Stunner instrument software.

## Protein mass determination by a mass photometer

Mass photometer (MP) experiments were performed using the Two$^{MP}$ mass photometer (Refeyn Ltd, Oxford, UK). The sample was added to the sample carrier slide (Refeyn) and silicone gaskets (6–3.0 mm diameter ID, RD501078, Refeyn) were used to hold the sample drops. To find focus, 16 µl of buffer (20 mM HEPES, 100 mM NaCl, pH 7.5 supplemented with either 10 mM EDTA, or 2 mM CaCl$_2$) were added into a well, and the focal position was identified and locked using the autofocus function. Next, 4 µl of B-RCAL at 0.5 µg/mL (previously incubated with the buffer for 1 h at room temperature) were added, thoroughly mixed, and images were acquired for 70 s at 502 Hz. The data were analysed using the Discover$^{MP}$ software (version 2.3).

## Co-crystallization of calprotectin and linear peptide 3

Calprotectin was expressed as His$_6$-linker-S100A9-linker-S100A8 (B-RCAL) fusion protein and purified as described in ref. 21. For crystallization purposes, calprotectin was expressed as His$_6$-linker-S100A9-linker-S100A8 (B-RCAL) fusion protein with cysteines replaced by serines by site-directed mutagenesis to prevent disulfide bond formation. Prior to crystallization, the His-tag and the linker sequence between S100A9 and S100A8 were cleaved by 3C precision protease in a 1:100 molar ratio overnight at 4 °C. The cleaved protein was purified by size-exclusion chromatography on a Superdex 200 10/300 GL column (GE Healthcare) using a buffer containing 20 mM HEPES, pH 7.4, and 100 mM NaCl. The protein was concentrated to a final concentration of 11 mg/mL (440 µM) in 20 mM HEPES, 100 mM NaCl, pH 7.4 using a centrifugation device with a 3-kDa cut-off. Prior to crystallization, a 14-fold molar excess of linear peptide 3 in water buffer was added to the protein to reach a final peptide concentration of 6.2 mM, and CaCl$_2$ was added to reach a concentration of 1 mM.

Crystals of the recombinant calprotectin with the peptide were grown at 18 °C employing the sitting drop vapor diffusion technique. Screening of 6 × 96 crystallization conditions was done using an automated Mosquito crystal robot (SPT Labtech). Several conditions of the PACT Premier (Molecular Dimensions) plate yielded a crystal that appeared between days 3 and 5. The droplets contained 200 nL of protein solution and 200 L of precipitant solution and were equilibrated against 100 µL of precipitant solution in a 96-well plate. The best crystals grew in the condition containing 0.1 M MIB (sodium malonate dibasic monohydrate, imidazole, boric acid), pH 6.0, 25% (w/v), PEG 1500, as the precipitant solution. The crystal was transferred to a cryogenic solution (25% glycerol) and flash-frozen in liquid nitrogen.

## X-ray structure determination

Data for calprotectin crystals in complex with linear peptide 3 were collected at the beamline PXIII of the Swiss Light Source at the Paul Scherrer Institute (SLS, Villigen, Switzerland) at a wavelength of 1.0 Å. Raw data were processed with the program XDS (XDS Program Package)[31]. The structure was solved by molecular replacement using Phenix (version 1.19.2)[32] with the atomic coordinates of calprotectin (PDB ID: 4GGF) as a search model. The peptide was manually modeled into the extra density using Coot (version 0.9.4.1)[33]. The structures were completed by iterative refinement in Phenix and model building in Coot, achieving a final model at 1.85 Å resolution. One copy of calprotectin (S100A8 and S100A9) and one copy of peptide constitute the asymmetric unit, while the calprotectin tetramer is formed by symmetry.

After performing molecular replacement of the protein chains with the reported calprotectin crystal structure 4GGF with metals removed, we identified extra-density regions surrounded by amino acid side chains that indicate the presence of metal ions. In an iterative process, we placed calcium, sodium, nickel, or potassium ions in those positions and checked their suitability using a web-based software tool (https://cmm.minorlab.org), replaced those not fitting, and repeated the process until all of them gave acceptable parameters. For each calprotectin heterodimer, we found six sites that are most likely occupied by metals. Each monomer has one non-canonical N-terminus EF-hand domain and one canonical C-terminus

EF-hand domain. Two additional binding sites are created between the monomers by 6xHis or 3xHis-1xAsp. For all of these sites, we proposed the type of metal as follows: monomer S100A8 contains a Na(I) atom in the N-terminus site and a Ca(II) atom in the C-terminus. In monomer S100A9, both sites are occupied by Ca(II) atoms. The 6xHis-shared site is occupied by Ni(II), while the 3xHis-1xAsp shared site is occupied by K(I).

The data collection and refinement statistics are summarized in Supplementary Table 2. The atomic coordinates and the structure factors were deposited in the Protein Data Bank: 7QUV. Molecular graphic figures were generated using ChimeraX (version 1.5) retrieved from https://www.cgl.ucsf.edu/chimerax/.

## Sandwich ELISA

The wells of 96-well Immulon 4 HBX-Extra High Binding plates (Thermo Fisher Scientific) were coated by overnight incubation at 4 °C with 80 µL of a 0.5 µg/mL solution of anti-calprotectin rabbit polyclonal antibody in $Na_2HCO_3$ (0.2 M, pH 9.4) (purchased from BÜHLMANN Laboratories AG; type: 300135; lot number: 202104). The plates were rinsed four times with 200 µL of washing buffer (25 mM Tris, 150 mM NaCl, 2 mM $CaCl_2$, pH 7.4, with 0.05% [v/v] Tween-20) per well. All the wells were blocked by incubation with 300 µL of blocking buffer (25 mM Tris, 150 mM NaCl, pH 7.4, with 0.05% [v/v] Tween-20, and 3% [w/v] BSA) for 1 h at room temperature with shaking. After this time, the blocking buffer was removed and 80 µL of calprotectin was added to the dilution buffer (washing buffer with 1% [w/v] BSA) for 1 h at room temperature with shaking. The wells were washed four times with washing buffer. Biotinylated linear peptide 3 (50 nM final concentration) was pre-mixed with Strep-HRP (12.5 nM final concentration; Thermo Fisher Scientific) for 30 min and 80 µL were added to the wells for 2 h at room temperature with shaking. The solution was removed and the wells were washed six times with a washing buffer. A volume of 80 µL of TMB substrate (ready-to-use solution, Sigma-Aldrich) were added for 30 min, and the reaction was stopped with 40 µL of sulfuric acid (2 M). The absorbance was read at 450 nm in a microwell plate reader (Infinite M200Pro, Tecan).

## Preparation of peptide-gold nanoparticles

To 1 ml of streptavidin-coated gold nanoparticles (Strep-AuNPs, 40 nm diameter; CGSTV-0600, Arista Biologicals) that have an $OD_{530}$ of 10, linear biotinylated peptide 3 (biotinylated at the C-terminus) was added (1 µM). The amount of peptide added was estimated to be sufficient for occupying all biotin-binding sites. Excess biotin was added to block potentially free biotin-binding sites. Peptide 3-AuNPs were concentrated by centrifugation and resuspended in a conjugate resuspension buffer.

## Half-strip assay

The nitrocellulose membrane was stuck onto a sticky backing card together with an absorbent pad that overlapped the nitrocellulose membrane by 2 mm. Strips of 0.5 cm were cut using scissors. The polyclonal rabbit anti-calprotectin antibody (purchased from BÜHLMANN Laboratories AG; 0.5 mg/mL stock; type: 300135; lot number: 202104) was applied as a dot to the test region 1.9 cm distant from the bottom of the nitrocellulose membrane. Recombinant calprotectin (B-RCAL) was applied as a dot to the test region 1.5 cm distant from the bottom of the nitrocellulose membrane. The half-strip was put vertically into a well of a flat-bottom 96-well plate containing 100 µL chase buffer, containing peptide 3-AuNPs (AuNP concentration: $OD_{530} = 0.4$) and calprotectin. The half-strip was removed after 15 min. For quantitative analysis, the intensity of the dots were measured using Image J (version 1.54). Curves were plotted using GraphPad Prism (version 5).

## Lateral flow assay

Peptide 3-AuNPs were immobilized on the conjugate-release pad (CRP). The membrane was added to the backing card and at one end, the absorbent pad was placed to collect the excess reagents. At the other end, the CRP previously soaked with peptide 3-AuNPs (AuNP concentration: $OD_{530} = 10$) was placed and overlaid with the sample application pad. Strips of 0.5 cm were cut using scissors. Test (T) and control (C) dots were immobilized on the nitrocellulose membrane by pipetting 0.5 µL polyclonal rabbit anti-calprotectin antibody (purchased from BÜHLMANN Laboratories AG; 0.5 mg/mL stock; type: 300135; lot number: 202104), or 0.5 µL linear peptide 3-biotin pre-incubated with neutravidin, and 0.5 µL recombinant calprotectin (B-RCAL), respectively. Buffer spiked with known quantities of recombinant calprotectin (B-RCAL) was applied in volumes of 80 µL to the LF strips. For quantitative analysis, the intensity of the dots were measured after 15 minutes using Image J. Curves were plotted using GraphPad Prism.

## Binding of peptide to native calprotectin

Native calprotectin is derived from granulocytes (kind gift from Thomas Vogl, Institute of Immunology, University of Muenster) and was purified as described in ref. 34. The size and concentration was analyzed and compared to B-RCAL by standard SDS-PAGE. The affinity of peptide 3 for native calprotectin was measured in a fluorescence polarization assay as described above.

## Lateral flow assay cassettes

The membrane and pads were assembled as described above, cut into strips of 0.5 cm width, and assembled into cassette housings. Samples containing calprotectin were applied to the LF strips. Whole blood was anticoagulated with EDTA and spiked with a 9.5 µg/mL B-RCAL. Plasma was obtained by centrifugation of the calprotectin-spiked blood. The blood and plasma samples were diluted ten times in chase buffer. Patient samples were obtained as serum fractions and were diluted ten times with chase buffer before addition to the LF strip. To run the LFA, 80 µL of the sample was applied to the SAP. After the sample solution was allowed to migrate through the membrane for 15 min at room temperature, the appearance of red lines at C and T positions indicated the detection of calprotectin. A negative result was indicated by the appearance of a red line only on the C line. For a quantitative analysis, the intensity was measured using a Quantum Blue® reader (third generation). Curves were plotted using GraphPad Prism.

## Patient samples

Patient serum samples were purchased by BÜHLMANN Laboratories from In.Vent Diagnostica GmbH (Henningsdorf, Germany) who had received ethical approval from the Freiburg Ethics Committee International (feki). In.Vent Diagnostica GmbH had received consent from patients to use the samples for research.

## Reporting summary

Further information on research design is available in the Nature Portfolio Reporting Summary linked to this article.

# Data availability

Three Supplementary Tables and 14 Supplementary Figures are provided in the Supporting Information. Four movies showing the LFA assay are provided as Supporting Movies. Raw data are provided in a Source Data file. The atomic coordinates of linear peptide 3 bound to calprotectin is deposited in the PDB (https://www.rcsb.org) under the accession code 7QUV. The X-ray structure 4GGF used in this work and published before can be found in the PDB too. Source data are provided with this paper.

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

## Acknowledgements

This work was supported by an Innosuisse grant (28020.1 IP-LS) that was granted to EPFL (C.H.) and BÜHLMANN Laboratories (C.-B.G.) and sponsored the position of C.D.-P. We are grateful to Thomas Vogl from the Institute of Immunology, University of Münster, for providing native calprotectin. We thank Amédé Larabi from the EPFL PTPSP for help with the crystallization screen and local contacts at the Swiss Light Source (PSI, Villigen) for X-ray data collection.

## Author contributions

C.D.-P., C.H., and C.-B.G. conceived the phage display selection and peptide characterization experiments and C.D.-P. performed all experiments. B.R., C.D.-P., D.G., and C.-B.G. planned and performed the lateral flow assays. F.P., K.L., and C.D.-P determined the X-ray structure. L.F.-S. developed the Met-deficient peptide and analyzed the size of calprotectin. C.H. and C.D.-P. wrote the manuscript. All authors discussed the results and edited/commented on the manuscript.

## Competing interests

C.D.-P., B.R., L.F.S., D.G., F.P., K.L., C.-B.G., and C.H. are inventors of a patent application covering the herein-described peptidic calprotectin ligands and their application in diagnostic assays. B.R., D.G., and C.-B.G. are employees of BÜHLMANN Laboratories AG that sells immunoassays for calprotectin.
