## [Peer Review File · Nature Communications]

High-affinity peptides developed against calprotectin and their application as synthetic ligands in diagnostic assaysREVIEWER COMMENTS

Reviewer #1 (Remarks to the Author):

Cristina et al. developed a novel calprotectin (CP) lateral flow assay based on peptides. They show robust antibody free diagnostic assay specific for CP tetramer. This work is original and is of wide interest for inflammatory disease diagnostics. A larger focus has been given to optimize the LFA cassette to avoid discrepancies in current CP tests. Due to ease of synthesis, high stability, and accuracy, this study suggests that synthetic ligands can be used instead of antibodies. These experiments were conducted in an appropriate manner, and the methodology is adequate. However, since this manuscript uses a novel methodology, there are some concerns that require further explanation from the authors.

Comments and Questions,

1. Authors should discuss the residues which link S100A9 and S100A8.

2. The R_{work} -0.2205 and R_{free} -0.2754 values are high for 1.85Å structure. Is there any possibility to revisit the refinement? I suggest running composite omit map for the peptide and include the omit map in Supplementary Information.

3. In the Supplementary information, authors stated that there are 3 Ca^{2+} ions and three other metal ions in the CP. There are two separate Ca^{2+} binding sites in S100A8 and S100A9. At high concentrations of Ca^{2+} such as 2 mM, there should be 8 Ca^{2+} in a CP tetramer. It is necessary to properly assign the metals in CP or to provide a clear explanation. It is worth noting how authors assigned the Ni^{2+} ion.

4. It would be important to mention that the tetramer is formed by symmetry in the main text, even though it is noted in the Materials and Methods section.

5. In Fig. 4C the linear peptide 3-biotin has stronger signal, is there any clear reason for this result?

6. Even though 10 μ M Calprotectin should form tetramers at 2 mM Ca^{2+} concentration, it is important to confirm the tetramer formation using DLS at the least.

7. When authors determined the stability of peptide-AuNP by measuring UV-Vis, the supplementary Fig.9 shows lower intensity for 18th day which indicates aggregation or loss of AuNP particles. Is the difference significant between 14th and 18th day? If possible, I would like to see a triplicate since AuNP stability is important.

8. It was unclear if the streptavidin-coated gold nanoparticles were purchased or made in the laboratory. What is the concentration of streptavidin-coated gold nanoparticles? What is the AuNP size? If they are purchased, mention the vendor. Such details would be helpful if someone wants to reproduce the work.

Reviewer #2 (Remarks to the Author):

Thank you very much for the opportunity to gain insight into this interesting work.

This work addresses an important issue, i.e. that current methods for detecting calprotectin differ considerably which is often not recognized by researchers and clinicians (e.g. Jukic, Gut, 2021). This work describes the development and characterization of a peptide with specific binding to a heterotetrameric calprotectin complex. The structure of the complex was solved, demonstrating how the peptide

associates with the complex. A huge effort was put into the selection, analysis and testing of the selected peptide binding to the calprotectin complex. The peptide enabled detection of low calprotectin concentrations with high affinity. The peptide was shown to be applicable for LFA analysis detecting calprotectin in blood as demonstrated on serum samples from rheumatoid arthritis patients.

We have some proposals for the manuscript

- When reading this article we were not sure whether the LFA was intended for analysis of calprotectin in blood or in feces. We agree with the authors that most of current methods used in clinical practice to measure calprotectin levels from patients with IBD are performed on fecal samples. The development of a more precise method would strengthen the use of calprotectin analysis. The use of fecal samples for the measurement of calprotectin introduces a lot of variation due to the nature of the sample material and the collection of sample material, which is often handled by the patient. The development of a method to analyze calprotectin in blood samples could probably provide some advantages. The sampling of blood is much easier and the following analysis would most likely offer faster results making it possible to follow the treatment and disease development.

- The introduction emphasizes the need for a better method to detect calprotectin in IBD and the last sentence in the conclusion expresses an expectation that this peptide could provide such method. However, when testing the LFA analysis, serum samples from 18 rheumatoid arthritis patients have been used. Have you considered using the LFA analysis on IBD samples?

- Further, the introduction primarily deals with detecting calprotectin in blood. Would it be possible to include more about the relevance of the heterotetrameric calprotectin complex in blood? According article by Mortensen et. al.(Journal of Crohn's and Colitis, 2022) the heterotetrameric structure of calprotectin is rapidly degraded in blood.

- Have you considered comparing the results from the LFA analyses with results from existing methods?

- We missed a description of how native calprotectin was derived from granulocytes? Please, add a reference or description.

- We were not able to retrieve the article stated as reference 18. Please review the references and check availability.

This work could potentially be important to the field and advance understanding in a way that will move the field forward.

Our research group is based at a hospital and we have a clinical approach when reading this article.

Yours sincerely

Vibeke Andersen

Professor, Head of Research, spec in internal medicine and medical gastroenterology, MD, PhD,

Research unit of Molecular Diagnostics and Clinical Research, Medical Department, Institute of Regional Health Research, University Hospital of Southern Denmark

Reviewer #3 (Remarks to the Author):

I read the work submitted for review with great interest. It contains very extensive research material, and a logical sequence of consecutive experiments, ultimately leading to the development of a diagnostic test. This test uses a peptide instead of an antibody. This test is very specific and I generally like the idea very much. The publication uses various research techniques that have been selected in the right way.

I have a few questions and comments on the manuscript submitted for review:

Major:

1) HPLC chromatograms of most peptides have one major peak with a retention time of approx 11 min and very close with a slightly less retention time of approx 10 min a second peak. Do you know what contamination is hidden in this signal (the one with lower intensity and lower retention time)? What is the mass of this pollutant? What was the purity level of the peptides used in the research?

2) Why the KD was calculated using a 1:1 fitting. Why was 1:2 fitting not applied? (see Fig. 6S)

3) Verse 164 et seq. It says that "due to the presence of two identical calprotectin proteins forming a 2-fold symmetry, each calprotectin tetramer can bind two peptides". In my opinion, this statement should be supported by an additional experiment in which the authors would prove that the peptide binds to the dimer and that this dimer (peptide: dimer) together with the other dimer (peptide: dimer) forms a tetramer. It is true that the structure of the X-ray complex is shown (it is great that this experiment was successfully performed), but there is no confirmation that such a system is formed in a solution. For example, the SEC technique, electrophoresis, or DLS could be used.

4) There are differences in the values of the KD constants determined by the fluorescence anisotropy method and the SPR method. Only FP values are reported in the main manuscript. There is no discussion explaining the differences in the constant values between the two techniques. The SPR data indicate much higher values of these constants.

5) The use of a peptide with a Met residue in its sequence in diagnostic tests may be (or may not be) somewhat risky due to the fact that methionine may be oxidized. This would mean that the peptide (and therefore the entire diagnostic test) should have a specific expiry date. Have the authors investigated the susceptibility of Peptide 3 to the oxidation process, and more specifically methionine residues? Will such a peptide still bind with similar strength to calprotectin?

6) I have a question about the "Preparation of peptide-gold nanoparticles" procedure. Are the authors able to assess the peptide concentration on the surface of AuNPs? Since the peptide binds to streptavidin, the more appropriate question seems to be: is it possible to estimate the concentration of streptavidin on the surface of gold nanoparticles, and how? Some materials and methods lack information on how streptavidin nanoparticles were prepared.

7) In my opinion the quality of the gel in Fig. 11S could be better. There is no description of the gel. I found information that the human homologue of calprotectin dimer is 24 kDa. So you can see the B-RCAL protein dimer and the native protein monomer on the gel? Why doesn't native protein form a dimer?

Minor:

1) In the introduction, the authors explain that there are tests based on antibodies on the market that are used to identify calprotectin. It would be nice to mention the names of these products.

2) Usually the constant Kd is marked with a capital letter "D", that is KD. → What does the KdS symbol mean?

3) In Fig. 1, reference numerals appear in red font. What do they mean? This should be explained in the figure caption.

4) Fig. 5S. Why are there no standard deviations marked for peptides 9-13?

5) "SPR sonograms" should be "SPR sensorgrams"

6) Calcium ions should be marked as Ca²⁺ or Ca (II) and in the text. There should be calcium ions or calcium chloride, not calcium (eg, verse 151).

7) The title of the subsection says "Peptides bind in" dumbbell "and linear configuration". In my opinion, the word configuration should be replaced with the word conformation because the configuration is a concept that describes spatial orientation with regard to the chirality of atoms.

8) Can the different degrees of loading of AuNPs nanoparticles with the peptide affect the result of the diagnostic test?

Point-by-point discussion of reviewer reports and changes

We would like to thank the reviewers for their valuable input and contributions to improve our manuscript. Below is a point-by-point response to their comments and a detailed description of the revisions made in response to their suggestions. We provide the revised manuscript and supplementary information with all the changes highlighted in yellow.

REVIEWER COMMENTS

Reviewer #1 (Remarks to the Author):

Cristina et al. developed a novel calprotectin (CP) lateral flow assay based on peptides. They show robust antibody free diagnostic assay specific for CP tetramer. This work is original and is of wide interest for inflammatory disease diagnostics. A larger focus has been given to optimize the LFA cassette to avoid discrepancies in current CP tests. Due to ease of synthesis, high stability, and accuracy, this study suggests that synthetic ligands can be used instead of antibodies. These experiments were conducted in an appropriate manner, and the methodology is adequate. However, since this manuscript uses a novel methodology, there are some concerns that require further explanation from the authors.

Comments and Questions,

1. Authors should discuss the residues which link S100A9 and S100A8.

Our answer: We describe now better the linker between S100A9 and S100A8 of calprotectin. Specifically, we indicate that all 8 amino acids of the inserted linker belong to the HRV 3C protease recognition site.

Changes (in yellow):

*We screened against recombinantly expressed calprotectin B-RCAL (BÜHLMANN recombinant calprotectin; His6-linker-S100A9-linker-S100A8) that fused the S100A9 and S100A8 proteins via a cleavable linker (LEVLFQ/GP; **all eight amino acids of the linker belong to the HRV 3C protease recognition site; cleavage site indicated**) and that carried an N-terminal His-tag appended via a second cleavable linker (MM[H]6LEVLFQ/GP)¹⁸. Because B-RCAL has the same structure and essentially the same immunological properties as native calprotectin purified from granulocytes¹⁸, we used the recombinant protein without removing the His-tag or cleaving **the 8-amino acid** linker between the two proteins for all phage display selections and the initial peptide characterization experiments.*

2. The Rwork-0.2205 and Rfree-0.2754 values are high for 1.85Å structure. Is there any possibility to revisit the refinement? I suggest running composite omit map for the peptide and include the omit map in Supplementary Information.

Our answer: The Rwork and Rfree have indeed high values. Despite several attempts and rounds of refinement, we were not able to reduce them. We followed the suggestion of the reviewer and run an omit map which nicely showed that the electron density for the ligand comes from the data and not from

the model. We added the following text to the manuscript and the omit map to the new Supplementary Fig. 9 (the numbering of the following SI Figures was adjusted):

New sentence in manuscript:

An omit map in the absence of the modelled peptide unambiguously indicated that the positive features of the electron density came from the peptide (Supplementary Fig. 9).

3. In the Supplementary information, authors stated that there are 3 Ca²⁺ ions and three other metal ions in the CP. There are two separate Ca²⁺ binding sites in S100A8 and S100A9. At high concentrations of Ca²⁺ such as 2 mM, there should be 8 Ca²⁺ in a CP tetramer. It is necessary to properly assign the metals in CP or to provide a clear explanation. It is worth noting how authors assigned the Ni²⁺ ion.

Our answer: We have revisited the ions in the calprotectin structure to make sure that all is correct. We have added the following information to the materials and methods section to explain how we have identified the ion binding sites and the type of ions.

New information in the materials and methods section:

After performing molecular replacement of the protein chains with the reported calprotectin crystal structure 4GGF with metals removed, we identified extra density regions surrounded by amino acid side chains that indicate the presence of metal ions. In an iterative process, we placed calcium, sodium, nickel or potassium ions in those positions and checked their suitability using the web-based software tool below, replaced those not fitting and repeated the process until all of them gave acceptable parameters.

Webserver: <https://cmm.minorlab.org>

For each calprotectin heterodimer, we found 6 sites that are most likely occupied by metals. Each monomer has one non-canonical N-terminus EF-hand domain and one canonical C-terminus EF-hand domain. Two additional binding sites are created between the monomers by 6xHis or 3xHis-1xAsp. For all of these sites, we proposed the type of metal as follows: monomer S100A8 contains a Na(I) atom in the N-terminus site and a Ca(II) atom in the C-terminus. In monomer S100A9, both sites are occupied by Ca(II) atoms. The 6xHis-shared site is occupied with Ni(II) while the 3xHis-1xAsp shared site is occupied by K(I).

4. It would be important to mention that the tetramer is formed by symmetry in the main text, even though it is noted in the Materials and Methods section.

Our answer: Done

Sentence added:

One copy of calprotectin (S100A8 and S100A9) and one copy of peptide constitute the asymmetric unit, while the calprotectin tetramer is formed by symmetry.

5. In Fig. 4C the linear peptide 3-biotin has stronger signal, is there any clear reason for this result?

Our answer: We did not find an obvious reason for the different performance of the two peptides in ELISA. Given that i) both termini are free in the X-ray structure and ii) there are rather long linkers between peptide and biotin (Gly-Ser-Gly-Ser-Gly-Ser), we expected that the two peptide formats would bind similarly. We added text to describe our expectation and that we do not know the reason for the difference.

Changes (in yellow):

Linear Peptide 3 biotinylated at the C-terminus (linear Peptide 3-biotin) gave slightly stronger signals than the peptide immobilized via the N-terminus (biotin-linear Peptide 3) for reasons we could not rationalize based the X-ray structure and the good accessibility of both peptide ends. Given the better performance of the C-terminally conjugated peptide, we used the linear Peptide 3-biotin for further experiments.

6. Even though 10uM Calprotectin should form tetramers at 2 mM Ca²⁺ concentration, it is important to confirm the tetramer formation using DLS at the least.

Our answer: We performed this additional dynamic light scattering (DLS) experiment proposed by the reviewer 1. In addition, we compared the mass of the two calprotectin forms using a new instrument we have available since recently (mass photometry). Both experiments showed that calprotectin forms the expected tetramer in presence of calcium ions, and a dimer in absence of calcium ions. The data is shown in the new Supplementary Fig. 7.

New text added to manuscript:

Analysis by dynamic light scattering (DLS) and mass photometry (MP), that measure the hydrodynamic radius and mass of proteins, respectively, assured that the calprotectin applied indeed occurred as tetrameric structure in presence of calcium ions, and as dimer in absence of calcium ions (Supplementary Fig. 7).

Legend new Supplementary Fig. 7:

Supplementary Fig. 7. Analysis of the oligomeric state distribution of calprotectin in presence and absence of calcium ions. B-RCAL in buffer containing CaCl₂ or EDTA was analyzed by mass photometry (MP) (a) and dynamic light scattering (DLS) (b). EDTA was used to capture Ca²⁺ already present in the buffer of stock protein. The expected molecular weight and radius of B-RCAL dimer are 26.9 Da and 5.71 nm, respectively. The expected molecular weight and radius of B-RCAL tetramer are 53.8 Da and 6.82 nm, respectively. The molecular weights and hydrodynamic radii found by DLS and MP indicate that calprotectin forms a tetramer in presence of CaCl₂ and a dimer in absence (EDTA).

7. When authors determined the stability of peptide-AuNP by measuring UV-Vis, the supplementary Fig. 9 shows lower intensity for 18th day which indicates aggregation or loss of AuNP particles. Is the difference significant between 14th and 18th day? If possible, I would like to see a triplicate since AuNP stability is important.

Our answer: The variation seen in Supplementary Fig. 9 (Supplementary Fig. 11 now due to the addition of two SI Figures) is rather close or within the error range of the method used, and does not necessarily indicate a stability issue (but it could, as indicated by the reviewer). In order to assess the stability of Peptide 3 AuNPs over a long period, we compared LFA cassettes that were immediately used after assembly, with LFA cassettes that were stored for 11 months (derived from the same production lot). We found essentially the same performance and band intensities for the two experiments. This is not a systematic study of the stability but suggests that the Peptide 3-based LFA cassettes can be stored for a long time. We have added this information to the manuscript as follows.

New information at the end of the results section:

While we did not yet assess the stability and storage requirements for the Peptide 3-AuNP-based LFA in a systematic study, we found that cassettes that were stored for 11 months in dry atmosphere detected the same concentrations of calprotectin, hinting to a good stability (Supplementary Fig. 14).

8. It was unclear if the streptavidin-coated gold nanoparticles were purchased or made in the laboratory. What is the concentration of streptavidin-coated gold nanoparticles? What is the AuNP size? If they are purchased, mention the vendor. Such details would be helpful if someone wants to reproduce the work.

Our answer: We indeed did not provide sufficient details about the gold nanoparticles use. The streptavidin-coated gold nanoparticles are from Arista Biologicals (product name and number: Steptavidin CGC; CGSTV-0600). We have added this information as well as the particle size and the concentrations used in the different experiments.

Changes:

To 1 ml of streptavidin-coated gold nanoparticles (Strep-AuNPs, 40 nm diameter; CGSTV-0600, Arista Biologicals) that have an OD₅₃₀ of 10, linear biotinylated Peptide 3 (biotinylated at the C-terminus) was added (1 μM). The amount of peptide added was estimated to be sufficient for occupying all biotin-binding sites. Excess of biotin was added to block potentially free biotin-binding sites.

... containing Peptide 3-AuNPs (AuNP concentration: OD₅₃₀ = 0.4) and ...

... soaked with Peptide 3-AuNPs (AuNP concentration: OD₅₃₀ = 10) was placed ...

Reviewer #2 (Remarks to the Author):

Thank you very much for the opportunity to gain insight into this interesting work.

This work addresses an important issue, i.e. that current methods for detecting calprotectin differ considerably which is often not recognized by researchers and clinicians (e.g. Jukic, Gut, 2021). This work describes the development and characterization of a peptide with specific binding to a heterotetrameric calprotectin complex. The structure of the complex was solved, demonstrating how the peptide associates with the complex. A huge effort was put into the selection, analysis and testing of the

selected peptide binding to the calprotectin complex. The peptide enabled detection of low calprotectin concentrations with high affinity. The peptide was shown to be applicable for LFA analysis detecting calprotectin in blood as demonstrated on serum samples from rheumatoid arthritis patients.

We have some proposals for the manuscript

- When reading this article we were not sure whether the LFA was intended for analysis of calprotectin in blood or in feces. We agree with the authors that most of current methods used in clinical practice to measure calprotectin levels from patients with IBD are performed on fecal samples. The development of a more precise method would strengthen the use of calprotectin analysis. The use of fecal samples for the measurement of calprotectin introduces a lot of variation due to the nature of the sample material and the collection of sample material, which is often handled by the patient. The development of a method to analyze calprotectin in blood samples could probably provide some advantages. The sampling of blood is much easier and the following analysis would most likely offer faster results making it possible to follow the treatment and disease development.

Our answer: This is a good question and we agree with all the considerations and remarks made by the reviewer. We like to apply the peptide-based LFA for the quantification of calprotectin in both, feces and in blood samples. We now make this clearer in the last sentence of the conclusion section.

Modified sentence (new part in yellow):

We expect that these peptides will provide a simple, accurate, and non-invasive test for quantifying calprotectin in fecal or blood samples, for diagnosing and monitoring IBDs and other diseases.

- The introduction emphasizes the need for a better method to detect calprotectin in IBD and the last sentence in the conclusion expresses an expectation that this peptide could provide such method. However, when testing the LFA analysis, serum samples from 18 rheumatoid arthritis patients have been used. Have you considered using the LFA analysis on IBD samples?

Our answer: This is a good point. The application to RA and serum patient samples could indeed be confusing, given the introduction that focuses mainly on IBD and calprotectin quantification in feces samples. We used the serum RA samples due to the availability of the patient samples and our interest in quantifying calprotectin in blood (which is much less established compared to fecal calprotectin). We have now added this information as follows.

Sentence added:

We finally applied the LFA cassettes for quantifying calprotectin in patient samples, focusing on serum rather than feces samples due to their availability and the interest in assessing the compatibility of the peptide-based LFA with serum components.

- Further, the introduction primarily deals with detecting calprotectin in blood. Would it be possible to include more about the relevance of the heterotetrameric calprotectin complex in blood? According

article by Mortensen et. al.(Journal of Crohn's and Colitis, 2022) the heterotetrameric structure of calprotectin is rapidly degraded in blood.

Our answer: This is a good point. We added information and references to recent studies that report different oligomeric forms of calprotectin and metabolic products.

Additional sentence:

Quantification of specific oligomeric states or metabolic products of calprotectin, as observed in blood, might offer additional value for the diagnosis of diseases⁸⁻¹⁰.

- Have you considered comparing the results from the LFA analyses with results from existing methods?

Our answer: This is a step that is particularly important towards the development and positioning of a commercial product, but we have not yet done this.

- We missed a description of how native calprotectin was derived from granulocytes? Please, add a reference or description.

Our answer: The native calprotectin was kindly provided by Prof. Thomas Vogl (Univ. of Münster), who had purified it from granulocytes. We indicated this before in the acknowledgements and now describe it also in the materials and methods. We also provide a reference to the purification procedure.

Information added to materials and methods section:

Native calprotectin is derived from granulocytes (kind gift from Thomas Vogl, Institute of Immunology, University of Muenster) and was purified as described before³⁴.

- We were not able to retrieve the article stated as reference 18. Please review the references and check availability.

Our answer: The information for this reference 18 (now reference 21) was indeed not sufficient to find. We have now added all bibliographic details.

More detailed reference:

21. Gerhold, C.-B., Gerspach, M. A., Guschin, D., Takacs, M. & Weber, J. Recombinant calprotectin. Pat. Appl. WO20211706, BÜHLMANN LABORATORIES AG (2020).

This work could potentially be important to the field and advance understanding in a way that will move the field forward.

Our research group is based at a hospital and we have a clinical approach when reading this article.

Reviewer #3 (Remarks to the Author):

I read the work submitted for review with great interest. It contains very extensive research material, and a logical sequence of consecutive experiments, ultimately leading to the development of a diagnostic test. This test uses a peptide instead of an antibody. This test is very specific and I generally like the idea very much. The publication uses various research techniques that have been selected in the right way.

I have a few questions and comments on the manuscript submitted for review:

Major:

1) HPLC chromatograms of most peptides have one major peak with a retention time of approx 11 min and very close with a slightly less retention time of approx 10 min a second peak. Do you know what contamination is hidden in this signal (the one with lower intensity and lower retention time)? What is the mass of this pollutant? What was the purity level of the peptides used in the research?

Our answer: We have analyze this small peak for several peptides and found that it is peptide with oxidized methionine. The purity of the vast majority of peptides is greater than 90% (with some exceptions of less important peptides such as peptides 6 and 13). As described below, we have identified a variant of Peptide 3 that does not contain Met (Met10Leu). We have added the information about the impurity as well as the new methionine-deficient mutant (please see below).

New information added (figure legend Supplementary Fig. 4):

The small peaks eluting around one minute before the desired peptides, observed for many of the Peptide 3 variants, were identified to be peptide with oxidized methionine.

2) Why the KD was calculated using a 1:1 fitting. Why was 1:2 fitting not applied? (see Fig. 6S)

Our answer: In the SPR experiments, we had immobilized the calprotectin (tetramer) on the chip and analyzed binding of the peptides. In this way, the peptides bind independently to the two sites of a calprotectin tetramer and we thus fitted the data using a 1:1 binding model. This would be different if the peptide was immobilized on the chip and calprotectin could bind to two peptides (which would require analysis with a 1:2 binding model).

3) Verse 164 et seq. It says that “due to the presence of two identical calprotectin proteins forming a 2-fold symmetry, each calprotectin tetramer can bind two peptides”. In my opinion, this statement should be supported by an additional experiment in which the authors would prove that the peptide binds to the dimer and that this dimer (peptide: dimer) together with the other dimer (peptide: dimer) forms a tetramer. It is true that the structure of the X-ray complex is shown (it is great that this experiment was successfully performed), but there is no confirmation that such a system is formed in a solution. For example, the SEC technique, electrophoresis, or DLS could be used.

Our answer: The Peptide 3 binds only to calprotectin tetramer but not to dimer, hindering us to perform the proposed experiment. However, as also requested by reviewer 1, we have analyzed the oligomeric

state of calprotectin in presence and absence of calcium ions by DLS and mass photometry (MP), to ensure that we had worked truly with dimer and tetramer.

New text added to manuscript:

Analysis by dynamic light scattering (DLS) and mass photometry (MP), that measure the hydrodynamic radius and mass of proteins, respectively, assured that the calprotectin applied indeed occurred as tetrameric structure in presence of calcium ions, and as dimer in absence of calcium ions (Supplementary Fig. 7).

Legend new Supplementary Fig. 7:

Supplementary Fig. 7. Analysis of the oligomeric state distribution of calprotectin in presence and absence of calcium ions. B-RCAL in buffer containing CaCl₂ or EDTA was analyzed by mass photometry (MP) (a) and dynamic light scattering (DLS) (b). EDTA was used to capture Ca²⁺ already present in the buffer of stock protein. The expected molecular weight and radius of B-RCAL dimer are 26.9 Da and 5.71 nm, respectively. The expected molecular weight and radius of B-RCAL tetramer are 53.8 Da and 6.82 nm, respectively. The molecular weights and hydrodynamic radii found by DLS and MP indicate that calprotectin forms a tetramer in presence of CaCl₂ and a dimer in absence (EDTA).

4) There are differences in the values of the KD constants determined by the fluorescence anisotropy method and the SPR method. Only FP values are reported in the main manuscript. There is no discussion explaining the differences in the constant values between the two techniques. The SPR data indicate much higher values of these constants.

Our answer: All Kd values measured by SPR were indeed higher than those measured with the FP assay, for reasons we did not understand. It could be that the binding in solution (FP) versus binding to surface (SPR) causes the difference, but we are not sure.

We found some large differences for peptides in the initial "screening" phase (e.g. bicyclic peptide 3: 23 nM by FP versus 340 nM by SPR) but the differences were much smaller for peptides that were characterized in more detail (e.g. in triplicate), such as the isomer 1 of Peptide 3 (24 nM by FP versus 51 nM by SPR) or the linear Peptide 3 (25.5 nM by FP versus 46 nM by SPR).

We now describe better the Kd differences found for the two orthogonal methods and speculate about the reasons:

Changes (new text in yellow):

Regardless, the nanomolar binding affinity was confirmed by surface plasmon resonance (SPR) as an orthogonal binding assay, wherein the affinity measured by SPR was substantially weaker for some of the peptides (Supplementary Fig. 6a).

We did not find out why the affinity measurements by SPR gave weaker affinities, but could imagine that it is due the binding measurement in solution (FP) versus on a surface (SPR).

5) The use of a peptide with a Met residue in its sequence in diagnostic tests may be (or may not be) somewhat risky due to the fact that methionine may be oxidized. This would mean that the peptide (and therefore the entire diagnostic test) should have a specific expiry date. Have the authors investigated the susceptibility of Peptide 3 to the oxidation process, and more specifically methionine residues? Will such a peptide still bind with similar strength to calprotectin?

Our answer: Based on this feedback by the reviewers, we found that the methionine in Peptide 3 is indeed prone to oxidation. We tested alternatives to Met and found that peptide with Leu instead of Met bind equally well. We have added this new data to the manuscript.

New text:

A potentially vulnerable site of Peptide 3 is the methionine residue that can oxidize, as seen for some of the peptide variants synthesized (Supplementary Fig. 4; small peak eluting around one minute before the peptide). We tested several mutants of Peptide 3 and found that substitution of the Met10 to leucine conserves the binding affinity ($K_d = 30.6 \pm 4.6$ nM; Supplementary Fig. 5). In contrast to Peptide 3 in which a small quantity was oxidized upon storage in solution, the Met10Leu mutant did not show any oxidation at all, suggesting that a calprotectin LFA may be developed that can be stored for a very long time.

6) I have a question about the "Preparation of peptide-gold nanoparticles" procedure. Are the authors able to assess the peptide concentration on the surface of AuNPs? Since the peptide binds to streptavidin, the more appropriate question seems to be: is it possible to estimate the concentration of streptavidin on the surface of gold nanoparticles, and how? Some materials and methods lack information on how streptavidin nanoparticles were prepared.

Our answer: As also recommended by reviewer 1, we have added detailed information about the gold nanoparticles use. Specifically, we now indicate that the streptavidin-coated gold nanoparticles used are from Arista Biologicals (product name and number: Steptavidin CGC; CGSTV-0600) and we have added information about the particle size and the concentrations used in the different experiments. The density of streptavidin on the beads is not indicated by the commercial provider, but we assume that we occupy all biotin-binding sites with a peptide. Based on a related product offered by the same provider (antibody coated beads), we estimated that there are around 200 streptavidin proteins per gold nanoparticle.

Changes:

To 1 ml of streptavidin-coated gold nanoparticles (Strep-AuNPs, 40 nm diameter; CGSTV-0600, Arista Biologicals) that have an OD₅₃₀ of 10, linear biotinylated Peptide 3 (biotinylated at the C-terminus) was added (1 μ M). The amount of peptide added was estimated to be sufficient for occupying all biotin-binding sites. Excess of biotin was added to block potentially free biotin-binding sites.

... containing Peptide 3-AuNPs (AuNP concentration: OD₅₃₀ = 0.4) and ...

... soaked with Peptide 3-AuNPs (AuNP concentration: OD₅₃₀ = 10) was placed ...

7) In my opinion the quality of the gel in Fig. 11S could be better. There is no description of the gel. I found information that the human homologue of calprotectin dimer is 24 kDa. So you can see the B-RCAL protein dimer and the native protein monomer on the gel? Why doesn't native protein form a dimer?

Our answer: We agree and have labeled the bands on the gel. In addition, we provide the original gel photo (without parts cropped) in the raw data file.

Changes: See modified Supplementary Figure (now Supplementary Fig. 13).

Minor:

1) In the introduction, the authors explain that there are tests based on antibodies on the market that are used to identify calprotectin. It would be nice to mention the names of these products.

Our answer: There is a wide range of antibody-based calprotectin assays that can easily be found in the internet. In order to not favor one product/provider over another, we chose to not indicate examples, but we can do this in case the reviewers 3 strongly recommends to do so.

2) Usually the constant K_d is marked with a capital letter "D", that is KD. – What does the K_d symbol mean?

Our answer: We tried to find which abbreviation (K_D or K_d) should be used but did not find a clear answer and also not a strong preference for one form. We have now ensured to use always the same form. We can change it if there is a preference from the journal.

3) In Fig. 1, reference numerals appear in red font. What do they mean? This should be explained in the figure caption.

Our answer: These were the "labjournal" names of the peptides and linkers. We had forgotten to remove them but have done this now.

Changes: Figure 1 without the names in red.

4) Fig. 5S. Why are there no standard deviations marked for peptides 9-13?

Our answer: These peptides showed only very weak binding at high calprotectin concentrations (high micromolar). We considered them as much less important than the peptides from Library 2 (nanomolar affinities) and thus did not repeat the measurements to obtain SDs. We now indicate this in the figure legend.

Sentence added:

For peptides from Libraries 1 and 3 that showed rather weak binding in the first measurement, the FP assay was not repeated.

5) "SPR sonograms" should be "SPR sensorgrams"

Our answer: Corrected

Changes: "sonograms" → "sensorgrams"

6) Calcium ions should be marked as Ca²⁺ or Ca (II) and in the text. There should be calcium ions or calcium chloride, not calcium (eg, verse 151).

Our answer: Good point. All corrected

Changes:

calcium → calcium ion/s or calcium chloride

7) The title of the subsection says "Peptides bind in" dumbbell "and linear configuration". In my opinion, the word configuration should be replaced with the word conformation because the configuration is a concept that describes spatial orientation with regard to the chirality of atoms.

Our answer: We think that "conformation" would not be ideal either as this would be molecules with spatial arrangements of atoms that can be interconverted, which is not the case for the dumbbell and linear forms of the peptides. We change the description as follows.

New subtitle:

*Peptides bind in "dumbbell" and linear **form***

8) Can the different degrees of loading of AuNPs nanoparticles with the peptide affect the result of the diagnostic test?

Our answer: We did not observe that variation of the bead amount or bead-peptide loading had a large impact on the LFA performance. However, we need also to say that we did not systematically study the impact of these two parameters. An advantage of peptides as affinity reagents is that they can be used in well-defined quality (purity) and quantity, which will allow using gold nanoparticles with well-defined peptide copy numbers.

REVIEWERS' COMMENTS

Reviewer #1 (Remarks to the Author):

I am pleased to see that the authors have addressed the concerns within a short period of time. The new experiments were well conducted, and the analysis was well performed.

I would like to make a minor suggestion regarding figure making. Authors have used ChimeraX to prepare the Supplementary Fig. 9. I suggest they use the same template for Fig. 3. Although it is not necessary to address it, I believe it will make the figure more appealing.

I fully support publication of this manuscript in Nature Communications.

Reviewer #2 (Remarks to the Author):

Thanks to the authors for their responses. We have no further comments.

Best

Karen Mai Møllegaard, PhD student

Vibeke Andersen, Research leader, Professor

Reviewer #3 (Remarks to the Author):

Thank you very much for all the responses of the authors of the publication. I am satisfied with the responses and all the changes made to the manuscript. My only comment concerns the word "sensograms" used for SPR measurements. In my opinion, the word should contain an additional letter r meaning "sensorgrams". However, I do not insist on this word and leave it to the editors of the journal Nature to decide.

In my opinion, the publication contains a number of basic studies that ultimately led to the development of a diagnostic test.

I recommend the manuscript for publication.

Reviewer #4 (Remarks to the Author):

In this manuscript, Cristina et al. described the development of a lateral flow assay based on a peptide derived from phage display peptide library screening to detect calprotectin level in the blood samples from the rheumatoid arthritis patients. The authors presented substantial data to prove their claim, and especially for the phage display screening, the characterization of peptide-calprotectin tetramer binding, the determination of the structure of the peptide complex with the calprotectin tetramer. All these efforts should be appreciated, and they also provide new insight on the diagnosis of inflammatory disorders. The authors claimed that the peptide is easy to synthesize, has high stability and is accurate for detection, thus it may be superior to antibody-based detection. The novelty of this manuscript also include that the peptide can bind to a calprotectin tetramer instead of dimer while other detection methods are uncertain regarding this aspect.

Overall, the experiments designed are sufficient to support the conclusions, and the development of a new method for calprotectin tetramer can provide some new information on diagnosis of calprotectin. Moreover, I believe the authors already addressed most of the concerns from the reviewers of first round of review. I have some additional comments on the manuscript:

1. As the authors indicated, in the healthy adults, the decal calprotectin concentration is 10-34 $\mu\text{g/g}$, while in the patient with mild inflammation and severe IBD, the concentration is 80-160 $\mu\text{g/g}$, and above 160 $\mu\text{g/g}$. Given the slight difference between the health individuals and the patients with mild inflammation and the high specificity of the kit, how will the author distinguish the false positive as in the health individuals there are also a significant level of calprotectin. Because this is critical to clinical application, and evidence may need to be provided.

2. The authors claimed the advantages of this peptide-based detection over antibody-based kits,

while none of these claims have been evidenced by a head-to-head comparison with antibody-detection.

3. As the Reviewer 2 requested, the fecal sample as a non-invasive method, would be valuable for point-of-care and home detection. I don't know if the authors had already detected or not. If yes but not successful, I suggest present the data and discuss why it was not successful.

Minor:

1. How is the research advances of using phage display technology in calprotectin detection or even in detection of inflammatory disorders?

2. What is the limitation or improvement of the said method to discriminate different kinds of inflammatory disorders?

3. The peptides have several cysteines within the sequences, besides the intramolecular disulfide bond formation, are there any intermolecular connections to form peptide multimers? Is the peptide synthesis a reduction condition?

Point-by-point discussion of reviewer reports and changes

We would like to thank the reviewers for the feedback and the suggestion of a few last minor changes. Below is a point-by-point response to their comments and a detailed description of the revisions made. We provide the revised manuscript with the latest changes highlighted in yellow.

REVIEWERS' COMMENTS

Reviewer #1 (Remarks to the Author):

I am pleased to see that the authors have addressed the concerns within a short period of time. The new experiments were well conducted, and the analysis was well performed.

I would like to make a minor suggestion regarding figure making. Authors have used ChimeraX to prepare the Supplementary Fig. 9. I suggest they use the same template for Fig. 3. Although it is not necessary to address it, I believe it will make the figure more appealing.

I fully support publication of this manuscript in Nature Communications.

Our answer:

We have now used the software ChimeraX to present the structures in Figure 3. We agree that the figures made with this software look nicer.

Changes: See Figure 3 in manuscript

Reviewer #2 (Remarks to the Author):

Thanks to the authors for their responses. We have no further comments.

Best

Karen Mai Møllegaard, PhD student

Vibeke Andersen, Research leader, Professor

Reviewer #3 (Remarks to the Author):

Thank you very much for all the responses of the authors of the publication. I am satisfied with the responses and all the changes made to the manuscript. My only comment concerns the word "sensograms" used for SPR measurements. In my opinion, the word should contain an additional letter meaning "sensorgrams". However, I do not insist on this word and leave it to the editors of the journal Nature to decide.

Our answer:

Absolutely. We have corrected this.

Changes (4 × in SI): Sensogram → **Sensogram**

In my opinion, the publication contains a number of basic studies that ultimately led to the development of a diagnostic test.

I recommend the manuscript for publication.

Reviewer #4 (Remarks to the Author):

In this manuscript, Cristina et al. described the development of a lateral flow assay based on a peptide derived from phage display peptide library screening to detect calprotectin level in the blood samples from the rheumatoid arthritis patients. The authors presented substantial data to prove their claim, and especially for the phage display screening, the characterization of peptide- calprotectin tetramer binding, the determination of the structure of the peptide complex with the calprotectin tetramer. All these efforts should be appreciated, and they also provide new insight on the diagnosis of inflammatory disorders. The authors claimed that the peptide is easy to synthesize, has high stability and is accurate for detection, thus it may be superior to antibody-based detection. The novelty of this manuscript also include that the peptide can bind to a calprotectin tetramer instead of dimer while other detection methods are uncertain regarding this aspect.

Overall, the experiments designed are sufficient to support the conclusions, and the development of a new method for calprotectin tetramer can provide some new information on diagnosis of calprotectin. Moreover, I believe the authors already addressed most of the concerns from the reviewers of first round of review. I have some additional comments on the manuscript:

1. As the authors indicated, in the healthy adults, the fecal calprotectin concentration is 10-34 $\mu\text{g/g}$, while in the patient with mild inflammation and severe IBD, the concentration is 80-160 $\mu\text{g/g}$, and above 160 $\mu\text{g/g}$. Given the slight difference between the health individuals and the patients with mild inflammation and the high specificity of the kit, how will the author distinguish the false positive as in the health individuals there are also a significant level of calprotectin. Because this is critical to clinical application, and evidence may need to be provided.

Our answer:

This is a good point. The quantity of fecal calprotectin is indicative for the severity of IBD inflammation but not 100% predictive, as written by the reviewer. We have now changed the wording in the introduction to take this into account, by altering "correlates" by "shows a good correlation".

Changes (new part in yellow):

*The quantity of the marker calprotectin detected in feces **shows a good correlation** with endoscopic and histological data and indicates the severity of IBD1. In healthy adults, median levels of fecal calprotectin are 10–34 $\mu\text{g/g}$; values between 80–160 $\mu\text{g/g}$ can represent a mild organic disease or inflammation, and calprotectin above 160 $\mu\text{g/g}$ is highly indicative for active inflammation in the gut, wherein the values for the categorizations vary slightly in literature4–6.*

2. The authors claimed the advantages of this peptide-based detection over antibody-based kits, while none of these claims have been evidenced by a head-to-head comparison with antibody-detection.

Our answer:

Several advantages are intrinsic to peptides and are thus already provided, as for example the ease of production (chemical synthesis), the homogenous product (single species, no post-translational modifications), the high stability (no unfolding), and the site-directed immobilization.

The reviewer however likely pointed to a side-by-side comparison of peptide- and antibody-based LFAs. We have not yet performed an in depth side-by-side comparison. This will be important, in particular towards the development and positioning of a commercial product. We now state this in the conclusion section, to make this point clear.

Changes (new part in yellow):

Extensive studies will be required for side-by-side comparing the peptide-based LFA to existing antibody-based LFAs.

3. As the Reviewer 2 requested, the fecal sample as a non-invasive method, would be valuable for point-of-care and home detection. I don't know if the authors had already detected or not. If yes but not successful, I suggest present the data and discuss why it was not successful.

Our answer:

We have tested the peptide-based LFA for detecting calprotectin in blood samples, which is considered more difficult than detection in fecal samples. In fecal samples, calprotectin occurs at a higher concentration, which gives more leeway for dilution with buffer, allowing optimization of the conditions for the LFA. While we have not tested the peptide-based LFA with fecal samples, we expect that it will work without problems, for the reasons provided. We clarify that the new LFA format should be suited for detecting calprotectin in both, fecal samples and blood samples.

Changes:

We expect that these peptides will provide a simple, accurate, and non-invasive test for quantifying calprotectin in fecal or blood samples, for diagnosing and monitoring IBDs and other diseases.

Minor:

1. How is the research advances of using phage display technology in calprotectin detection or even in detection of inflammatory disorders?

Our answer:

Phage display technology may give rise to suitable binders to any kind of inflammatory marker that is used as antigen in biopanning experiments. Our successful example with calprotectin may encourage

others to apply phage display for developing diagnostic assays based on peptides. We now state this at the end of the conclusions section.

Changes:

In addition, the successful quantification of calprotectin with a phage display-selected peptide may stimulate the broader application of peptide-based affinity reagents for diagnostic assays.

2. What is the limitation or improvement of the said method to discriminate different kinds of inflammatory disorders?

Our answer:

A limitation is that it is slightly harder to develop peptide-based ligands (compared to antibodies). Advantages are the ease of synthesis, homogenous product, high stability, site-directed modification, and cheap production. We describe these strengths in the conclusions section.

Changes:

The robustness, high reproducibility, and ease of developing the assays observed herein can be attributed to a large extent to the ability of synthesizing the peptides in a homogenous, pure form and as conjugates for site-directed and convenient immobilization.

3. The peptides have several cysteines within the sequences, besides the intramolecular disulfide bond formation, are there any intermolecular connections to form peptide multimers? Is the peptide synthesis a reduction condition?

Our answer:

We had reduced the thiol groups before connecting cysteine pairs intra-molecularly by cyclization linkers. Multimers were not formed due to the full thiol reduction and the cyclization at low concentration (1 mM). We have clarified that cysteines were reduced before cyclization.

Changes (new part in yellow):

We synthesized the peptides with an N-terminal fluorescein label, cyclized random pairs of reduced cysteines by adding excess of cyclization reagents 1 to 6, and chromatographically separated the three isomers formed by bridging different cysteine pairs (Supplementary Fig. 4).